# LRRC15 inhibits SARS-CoV-2 cellular entry *in trans*

**Jaewon Song**[1ʘ], **Ryan D. Chow**[2ʘ], **Mario A. Peña-Hernández**[3,4ʘ], **Li Zhang**[1ʘ], **Skylar A. Loeb**[1], **Eui-Young So**[5], **Olin D. Liang**[5], **Ping Ren**[2], **Sidi Chen**[2], **Craig B. Wilen**[3,4]*, **Sanghyun Lee**[1]*

**1** Department of Molecular Microbiology and Immunology, Division of Biology and Medicine, Brown University, Providence, Rhode Island, United States of America, **2** Department of Genetics, Yale School of Medicine, New Haven, Connecticut, United States of America, **3** Department of Laboratory Medicine, Yale University, New Haven, Connecticut, United States of America, **4** Department of Immunobiology, Yale University, New Haven, Connecticut, United States of America, **5** Division of Hematology/Oncology, Department of Medicine, Rhode Island Hospital, Warren Alpert Medical School of Brown University, Providence, Rhode Island, United States of America

ʘ These authors contributed equally to this work.
* craig.wilen@yale.edu (CBW); sanghyun_lee@brown.edu (SL)

**Data Availability Statement:** All CRISPR screening data generated in this study are provided as S1 Data and are uploaded to NCBI's Sequence Read Archive (SRA) database under accession code

## Abstract

Severe Acute Respiratory Syndrome Coronavirus 2 (SARS-CoV-2) infection is mediated by the entry receptor angiotensin-converting enzyme 2 (ACE2). Although attachment factors and coreceptors facilitating entry are extensively studied, cellular entry factors inhibiting viral entry are largely unknown. Using a *surface*ome CRISPR activation screen, we identified human LRRC15 as an inhibitory attachment factor for SARS-CoV-2 entry. LRRC15 directly binds to the receptor-binding domain (RBD) of spike protein with a moderate affinity and inhibits spike-mediated entry. Analysis of human lung single-cell RNA sequencing dataset reveals that expression of LRRC15 is primarily detected in fibroblasts and particularly enriched in pathological fibroblasts in COVID-19 patients. *ACE2* and *LRRC15* are not coexpressed in the same cell types in the lung. Strikingly, expression of LRRC15 in ACE2-negative cells blocks spike-mediated viral entry in ACE2+ cell *in trans*, suggesting a protective role of LRRC15 in a physiological context. Therefore, LRRC15 represents an inhibitory attachment factor for SARS-CoV-2 that regulates viral entry *in trans*.

## Introduction

Severe Acute Respiratory Syndrome Coronavirus 2 (SARS-CoV-2) is the causative agent of Coronavirus Disease 2019 (COVID-19), representing a global health threat [1,2]. SARS-CoV-2 belongs to the β-coronavirus family along with Severe Acute Respiratory Syndrome Coronavirus (hereafter SARS-CoV-1) and Middle East Respiratory Syndrome Coronavirus (MERS-CoV) [3,4]. Like SARS-CoV-1, SARS-CoV-2 utilizes angiotensin-converting enzyme 2 (ACE2) as a receptor [5,6]. The viral structural protein spike (S), anchored on the surface of the viral envelope as homotrimers, binds to ACE2 and mediates virus entry [7]. The ectodomain of spike protein consists of the S1 and S2 subunits. The S1 subunit is comprised of the N-terminal

SRP349409. Analysis of bulk RNA-seq and scRNA-seq using previous publications were accessed directly from those publications.

**Funding:** This study was supported by NIH grants R00 AI141683 (S.L.), 2P20 GM109035-07 (S.L.), K08 AI128043 (C.B.W.), R01 AI148467 (C.B.W.), T32 GM007205 (R.D.C.), F30 CA250249 (R.D.C.) and P20 GM119943 (O.D.L.); the Smith Family Awards Program for Excellence in Biomedical Research (S.L.); a Burroughs Wellcome Fund Career Award for Medical Scientists (C.B.W.); the Ludwig Family Foundation (C.B.W.), the Mathers Charitable Foundation (C.B.W.); an Emergent Ventures fast grant (C.B.W.). DoD PRMRP IIAR (W81XWH-21-1-0019) (S.C.). The funders had no role in study design, data collection and analysis, decision to publish, or preparation of the manuscript.

**Competing interests:** I have read the journal's policy and the authors of this manuscript have the following competing interests: Yale (CBW) has a patent pending title "Compounds and Compositions for Treating, Ameliorating, and/or Preventing SARS-CoV-2 Infection and/or Complications Thereof."

**Abbreviations:** ACE2, angiotensin-converting enzyme 2; BMI, body mass index; COVID-19, Coronavirus Disease 2019; CRISPRa, CRISPR activation; dCas9, "dead" Cas9; DMEM, Dulbecco's Modified Eagle Medium; FBS, fetal bovine serum; GFP, green fluorescent protein; GTEx, Genotype-Tissue Expression; IgG, immunoglobulin G; ISG, interferon-stimulated gene; LRR, leucin-rich repeat; LRRC15, leucin-rich repeat-containing 15; MERS-CoV, Middle East Respiratory Syndrome Coronavirus; NTD, N-terminal domain; PAMP, pathogen-associated molecular pattern; RBD, receptor-binding domain; SARS-CoV-1, Severe Acute Respiratory Syndrome Coronavirus; SARS-CoV-2, Severe Acute Respiratory Syndrome Coronavirus 2; scRNA-seq, single-cell RNA-sequencing; sgRNA, single guide RNA; TMPRSS2, transmembrane protease serine 2; VSV, vesicular stomatitis virus.

domain (NTD) and the receptor-binding domain (RBD) [8]. The RBD of spike protein directly binds to ACE2, which induces a conformational change that facilitates virus fusion either with endosomal membrane or with the plasma membrane [6,9,10]. This fusion event releases the SARS-CoV-2 genome into the cytoplasm [11,12].

The interaction between the RBD of spike and ACE2 determines several key features of SARS-CoV-2 infection. The high affinity interface between the RBD and ACE2 is associated with higher infectivity of SARS-CoV-2 compared to SARS-CoV-1 [13], and a single point mutation at the RBD can alter host range and enable mouse infection [14–16]. Spike protein is the primary target antigen for COVID vaccines, and the majority of existing therapeutic antibodies function by blocking RBD and ACE2 interactions, indicating the importance of RBD and its binding to the cellular receptor for controlling SARS-CoV-2.

Thus far, several cellular factors have been identified to facilitate cellular entry of SARS-CoV-2. However, it is unclear whether there are any host factors that inhibit viral entry. Previous studies indicate that cleavage of spike protein by cellular proteases such as transmembrane protease serine 2 (TMPRSS2), cathepsins, and furin facilitates the entry of SARS-CoV-2 [9,11,17,18]. Several cellular surface proteins or glycans facilitate viral entry by acting as an attachment factor, which includes neuropilin-1 [19,20], heparan sulfate [21], and C-type lectins [22]. Alternative entry factors have been proposed such as AXL [23] and CD147 [24]. However, it remains to be elucidated whether any cellular entry factors regulate viral entry in a different manner.

In this study, we employed a screening method using the CRISPR activation (CRISPRa) technique. We generated a focused CRISPRa library, named *surface*ome, that covers all approximately 6,000 known/predicted surface proteins on the cellular plasma membrane. The *surface*ome screening with the SARS-CoV-2 spike protein revealed that human LRRC15 (leucin-rich repeat-containing 15) is a novel inhibitory attachment factor for SARS-CoV-2.

## Results

### A *surface*ome CRISPR activation screen identified cellular receptors for spike protein of SARS-CoV-2

To identify host factors that regulate SARS-CoV-2 entry, we performed the *surface*ome CRISPRa screen and investigated which cellular proteins regulate spike binding to cells. We specifically selected approximately 6,000 genes encoding plasma membrane proteins that contain either single or multiple transmembrane domains or are associated with the plasma membrane. We designed a CRISPRa library consisting of 4 activating single guide RNAs (sgRNAs) per gene and 1,000 nontargeting control sgRNAs (**S1A Fig**). The screen was performed in a human melanoma cell line, A375, as this cell line does not express endogenous *ACE2* and does not interact with SARS-CoV-2 spike protein without ectopic expression of *ACE2* [21]. A375 cells containing catalytically "dead" Cas9 (dCas9) were transduced with the sgRNA library and selected to produce a pool of cells with induced expression of individual surface proteins. We measured the binding of Fc-tagged S1 subunit of SARS-CoV-2 spike to the cells by flow cytometry. Cells exhibiting high fluorescent signal intensity were sorted and subjected to genomic DNA extraction and sgRNA sequencing (**Fig 1A** and **S1 Data**). Two biologically independent screen results indicated 2 only distinct hits, ACE2 and LRRC15 (**Fig 1B**). Other reported spike attachment factors were not significantly enriched in our screen [23,25,26] (**S1B Fig**). This discrepancy might be from relatively weak spike-binding affinities of previously identified attachment factors [23,26,27] or dependent on cell types expressing different cofactors. LRRC15 is a leucin-rich repeat domain-containing protein, which is an orphan cancer-associated protein [28,29]. There is no reported role of LRRC15 in SARS-CoV-2. An immunoglobulin G (IgG)

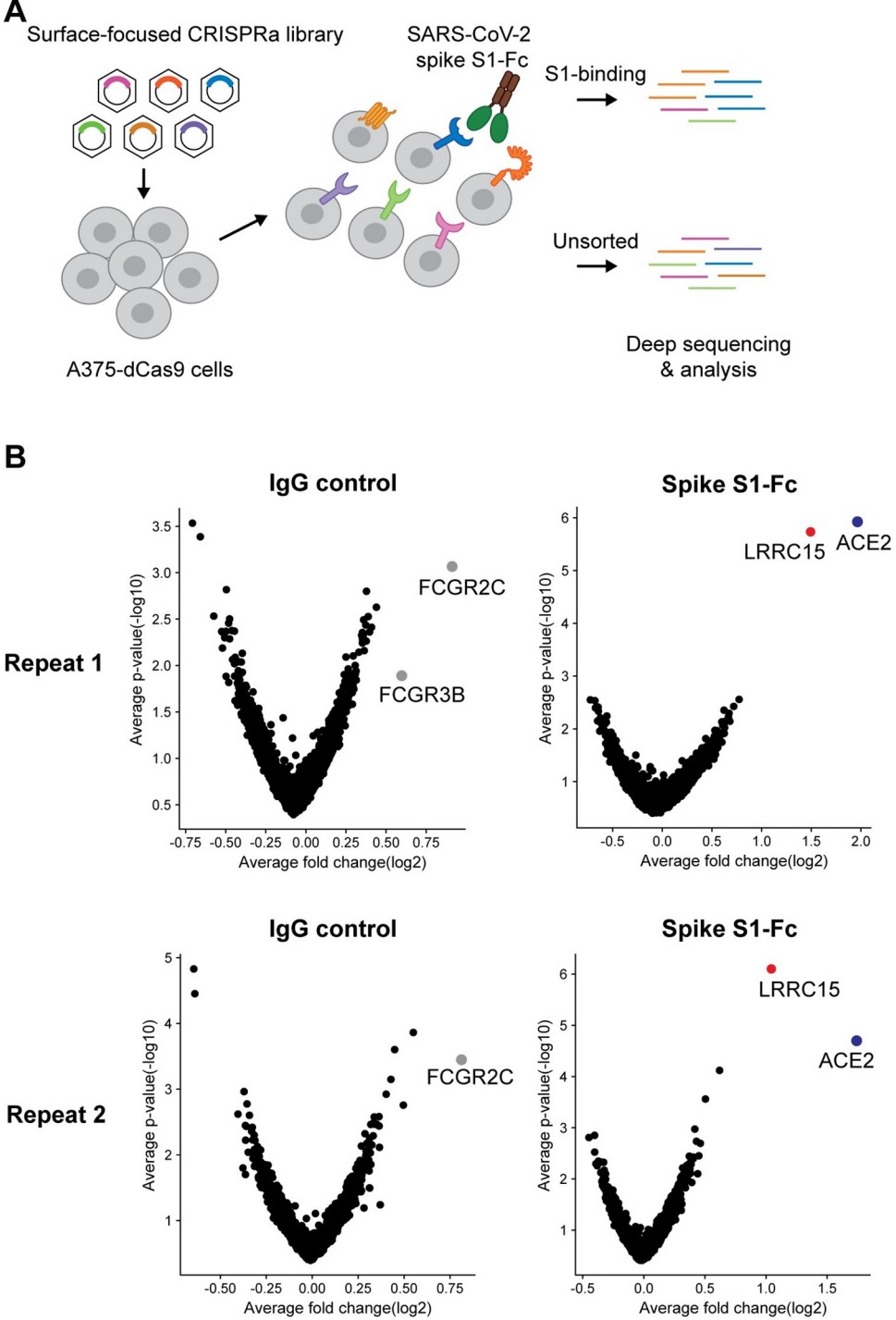

**Fig 1. A *surface*ome-focused CRISPRa screen identified cellular receptors binding with SARS-CoV-2 spike protein.** (**A**) Schematic of a focused CRISPRa screen for surface proteins interacting with SARS-CoV-2 spike S1-Fc fusion protein. (**B**) Volcano plots showing sgRNAs enriched or depleted in cells binding with SARS-CoV-2 spike S1-Fc or human IgG isotype control. Results from 2 biologically independent replicates are shown. For underlying data, see **S1 Data**. ACE2, angiotensin-converting enzyme 2; CRISPRa, CRISPR activation; dCas9, "dead" Cas9; IgG, immunoglobulin G; LRRC15, leucin-rich repeat-containing 15; SARS-CoV-2, Severe Acute Respiratory Syndrome Coronavirus 2; sgRNA, single guide RNA.

isotype control and anti-CD45 staining identified IgG receptor genes (*FCGR2C*, *FCGR3B*) and CD45-encoding gene, *PTPRC*, as the top hit, respectively, confirming that the *surface*ome CRISPR screening efficiently identifies cellular receptors for targeted proteins (**Figs 1B and S1C**).

## LRRC15 directly interacts with the spike via the receptor-binding domain

To validate the screening results, we utilized 2 different human cell lines, A375 and HeLa. These 2 cell lines do not express endogenous *ACE2* and are not susceptible to SARS-CoV-2 without ectopic expression of *ACE2* [21,30]. A375 and HeLa cells were transduced with 2 individual sgRNAs for *LRRC15* and a single sgRNA for *ACE2* to induce gene expression (**S2A and S2B Fig**). LRRC15-induced and ACE2-induced cells bound to the S1-Fc protein. The signal intensity in ACE2-induced cells was stronger than that of the LRRC15-induced cells. (**Fig 2A**). A similar pattern of protein interaction was observed in HeLa cells (**Figs 2B and S2B**). Trimeric full-length recombinant spike protein also bound to LRRC15-induced HeLa cells with the weaker signal intensity than ACE2-expressing cells (**S2C Fig**).

The interaction between LRRC15 and spike was further examined in a cell-free interaction model using recombinant proteins. An ELISA assay using recombinant LRRC15 and full-length spike indicated that LRRC15 directly interacts with the spike protein ($K_D$ = 109 nM). The affinity between LRRC15 and the spike seems to be weaker than that of ACE2 and spike (**Fig 2C**). Interaction with spike proteins of different SARS-CoV-2 variants was confirmed. Recombinant full-length spike proteins of $\alpha$ (B.1.1.7), $\beta$ (B.1.351), $\gamma$ (P.1), $\delta$ (B.1.617.2), and $\iota$ (B.1.526) variants were tested and LRRC15 interacted with all of these spike proteins with similar affinity (**S2D Fig**).

ACE2 interacts with the spike protein via the RBD but not the NTD [31]. Interestingly, we identified that LRRC15 interacts with the spike in a similar way. Interaction assays in cells and in a cell-free assay using ELISA indicated that the RBD is sufficient to recapitulate the interaction between LRRC15 and spike with a similar affinity compared to full-length S1 (**Fig 2D and 2E**). Next, we examined whether this interaction is specific to SARS-CoV-2 or conserved in other β coronaviruses. The ELISA assay using recombinant RBD protein of SARS-CoV-1 and MERS-CoV showed that LRRC15 binds to spike of SARS-CoV-1 with similar affinity but does not interact with spike of MERS-CoV (**S2E Fig**). These results indicate that LRRC15 is a novel cellular binding protein for the spike protein of SARS-CoV-1 and SARS-CoV-2 and directly interacts with the spike via the RBD.

## LRRC15 suppresses entry of SARS-CoV-2

We next investigated whether LRRC15 regulates the entry process of SARS-CoV-2. Pseudotyping a heterologous virus with spike protein has been utilized to study the entry process of SARS-CoV-2 [32,33]. To monitor viral entry, we utilized a replication-incompetent vesicular stomatitis virus (VSV) pseudovirus system that harbors the spike protein of SARS-CoV-2 and expresses a green fluorescent protein (GFP) reporter [34]. The LRRC15-induced A375 or HeLa cells were infected with the pseudovirus, and the infectivity was monitored by flow cytometry. Wild-type control A375 and HeLa cells were not susceptible to the spike-pseudotyped virus, consistent with previous reports [21,30] (**S3A and S3B Fig**). ACE2 but not LRRC15 expression was sufficient to support viral entry. The VSV-G-coated pseudovirus entered into all tested cell lines with a similar efficiency (**S3A and S3B Fig**). These data suggest that LRRC15 does not function as an entry receptor for SARS-CoV-2.

To examine whether LRRC15 regulates ACE2-mediated viral entry, we first generated a clonal HeLa cell line stably expressing ACE2, designated as HeLa-ACE2, and confirmed the

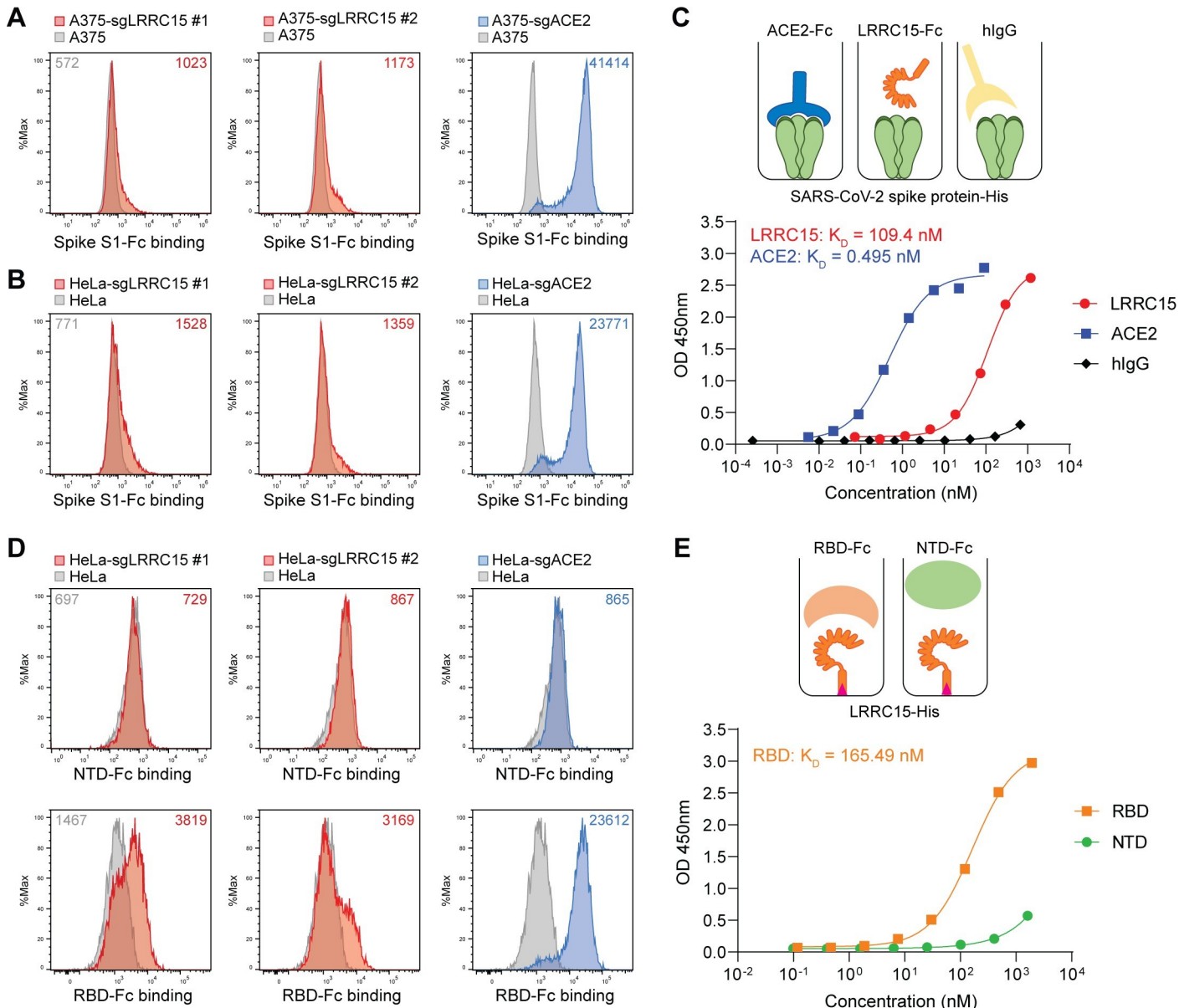

**Fig 2. LRRC15 binds with SARS-CoV-2 spike protein at the RBD.** (**A**) A375 cells were transduced with indicated activating sgRNAs and incubated with SARS-CoV-2 spike S1-Fc fusion protein. Protein binding was measured by flow cytometry with MFI shown. (**B**) HeLa cells were transduced with indicated activating sgRNAs and incubated with SARS-CoV-2 spike S1-Fc fusion protein. Protein binding was measured by flow cytometry with MFI shown. (**C**) Dose-dependent binding of SARS-CoV-2 spike protein (Wuhan-Hu-1) to both ACE2 and LRRC15 with an Fc tag was determined by ELISA. Human IgG1 was included as a negative control. Dots indicate means of duplicates. (**D**) HeLa cells were transduced with indicated activating sgRNAs and incubated with SARS-CoV-2 spike NTD-Fc or RBD-Fc fusion protein. Protein binding was measured by flow cytometry with MFI shown. (**E**) The binding of the SARS-CoV-2 RBD and NTD to LRRC15 was measured by ELISA. For underlying data, see **S3 Data**. ACE2, angiotensin-converting enzyme 2; LRRC15, leucin-rich repeat-containing 15; MFI, mean fluorescence intensity; NTD, N-terminal domain; RBD, receptor-binding domain; SARS-CoV-2, Severe Acute Respiratory Syndrome Coronavirus 2; sgRNA, single guide RNA.

high surface expression of ACE2 and susceptibility to SARS-CoV-2 spike pseudovirus (**S3C and S3D Fig**). Using this cell line, we induced LRRC15 expression with 2 different sgRNAs and measured the efficiency of pseudoviral entry. The increased LRRC15 expression resulted in a significant decrease in the spike-pseudotyped VSV entry, while the induction of an unrelated protein, CD45, showed similar infectivity compared to the mock control (**Figs 3A and S3E**). On the other hand, induction of DC-SIGN (CD209), an attachment factor that enhances

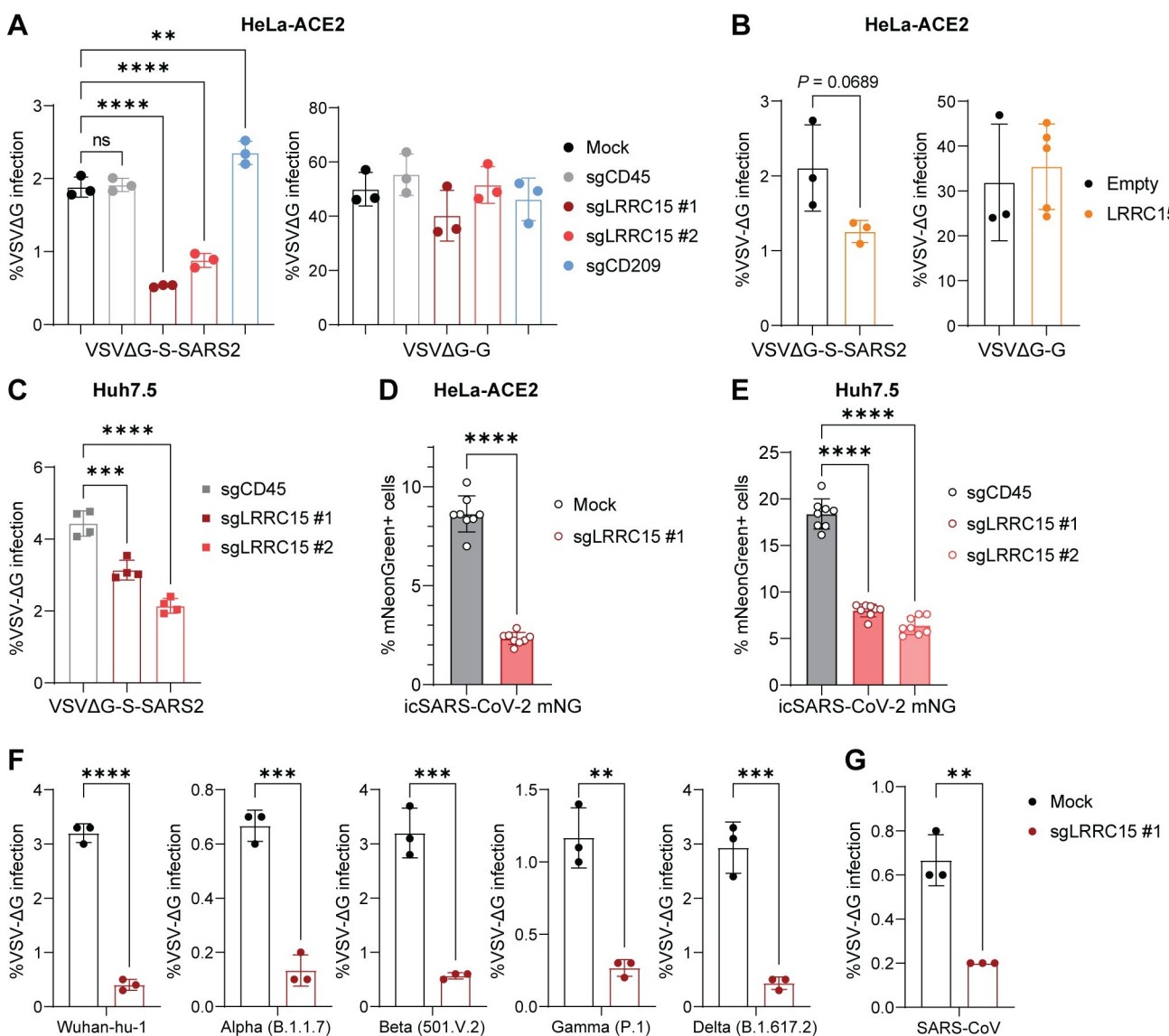

**Fig 3. LRRC15 inhibits ACE2-mediated SARS-CoV-2 entry.** (**A**) HeLa-ACE2 cells were transduced with indicated activating sgRNAs and infected with VSV pseudoviruses, VSVΔG-S-SARS2 or VSVΔG-G. GFP signal was measured at 20 hpi by flow cytometry ($n = 3$). Representative of 3 independent experiments are shown. (**B**) HeLa-ACE2 cells expressing LRRC15 or empty vector were infected with VSV psuedoviruses and GFP signal was measured at 20 hpi by flow cytometry ($n = 3$). Representative of 3 independent experiments are shown. (**C**) Huh7.5 cells were transduced with indicated activating sgRNAs and infected with VSVΔG-S-SARS2. GFP signal was measured at 20 hpi by flow cytometry ($n = 4$). Representative of 3 independent experiments are shown. (**D, E**) HeLa-ACE2 (D) or Huh7.5 (E) were transduced with indicated activating sgRNAs and infected with icSARS-CoV-2 mNG at 1 MOI. Infected cell frequencies were measured by mNeonGreen expression at 24 hpi ($n = 8$). Representative of 2 (**D**) or 3 (**E**) independent experiments are shown. (**F**) LRRC15-induced or mock HeLa-ACE2 cells were infected with VSV pseudoviruses harboring spike proteins of different SARS-CoV-2 variants. GFP signal was measured at 20 hpi by flow cytometry ($n = 3$). (**G**) LRRC15-induced or mock HeLa-ACE2 cells were infected with VSV pseudoviruses harboring SARS-CoV-1 spike. GFP signal was measured at 20 hpi by flow cytometry ($n = 3$). Data represent means ± SD (**A-G**). Data were analyzed by one-way ANOVA with Dunnett multiple comparisons test (**A, C, E**) or unpaired two-tailed t test (**B, D, F, G**). ns, not significant; $^*p < 0.05$; $^{**}p < 0.01$; $^{***}p < 0.001$; $^{****}p < 0.0001$. For underlying data, see **S3 Data**. ACE2, angiotensin-converting enzyme 2; GFP, green fluorescent protein; hpi, hours postinfection; LRRC15, leucin-rich repeat-containing 15; MOI, multiplicity of infection; SARS-CoV-1, Severe Acute Respiratory Syndrome Coronavirus; SARS-CoV-2, Severe Acute Respiratory Syndrome Coronavirus 2; sgRNA, single guide RNA; VSV, vesicular stomatitis virus.

ACE2-mediated viral entry, showed a significantly increased infectivity, confirming the specificity of the SARS-CoV-2 pseudovirus as a viral entry model [22,25]. The entry of VSV-G-pseudotyped virus was not affected by LRRC15 induction, indicating the LRRC15-mediated

inhibition is specific to the entry of SARS-CoV-2. The inhibitory function of LRRC15 was confirmed by ectopic overexpression of its cDNA as well (**Figs 3B and S3F**). Of note, a larger reduction in viral entry was observed with the sgRNA-mediated gene induction, likely due to the higher gene expression of LRRC15 on cells compared to the cDNA overexpression (**S3E and S3F Fig**). We employed another cell line, Huh7.5, which endogenously expresses ACE2 and is naturally susceptible to SARS-CoV-2 [35]. Inducing LRRC15 expression in Huh7.5 cells suppressed the entry of SARS-CoV-2 pseudovirus (**Figs 3C and S3G**). Next, we examined if LRRC15 is able to suppress the infection of live SARS-CoV-2. HeLa-ACE2 or Huh7.5 cells were infected with a full-length molecular clone of SARS-CoV-2, which expresses the mNeon-Green reporter, icSARS-CoV-2 WA01 mNG, and infectivity of the virus was measured by quantifying mNeonGreen fluorescence-positive cells [36]. SARS-CoV-2 infection was significantly inhibited by LRRC15 induction in both HeLa-ACE2 and Huh7.5 cells (**Fig 3D and 3E**). The suppressed viral entry was observed in other pseudotyped viruses containing the spike of multiple variants of SARS-CoV-2 (i.e., α, β, γ, and δ variants) and SARS-CoV-1 (**Fig 3F and 3G**), as expected by their similar binding affinities with LRRC15 (**S2C and S2D Fig**). Although LRRC15 reduced the entry of pseudotyped viruses of all tested variants at a similar fold in HeLa-ACE2 cells, different binding affinity may alter the phenotype in different settings like naturally susceptible cells or infection in the lung. In summary, these data indicate that LRRC15 suppresses spike-mediated viral entry, and the binding and suppression activity are specific to SARS-CoV-2 and SARS-CoV-1.

Cellular entry of SARS-CoV-2 is mediated by membrane fusion either at the plasma membrane or in the endosome. Cellular attachment factors and entry cofactors often differentially regulate the 2 cellular entry routes [37]. The cellular serine protease TMPRSS2 primes SARS-CoV-2 spike protein and facilitates viral entry through the plasma membrane route during infection [9]. Treatment with the TMPRSS2 inhibitor camostat in HeLa-ACE2 cells did not alter pseudoviral entry as these cells do not express endogenous TMPRSS2 [38], and the inhibitory effect of LRRC15 was conserved in camostat-treated cells (**S3H Fig**). We employed HEK293T cells ectopically expressing ACE2 and TMPRSS2 (293T-ACE2/TMPRSS2) to model the entry pathway through the plasma membrane fusion. In the 293T-ACE2/TMPRSS2 cells, camostat reduced entry of the SARS-CoV-2 pseudovirus. In 293T-ACE2/TMPRSS2 cells, LRRC15 suppressed viral entry only upon camostat treatment but not with control treatment (**S3I Fig**), implying that the TMPRSS2-independent SARS-CoV-2 entry is more sensitive to the entry restriction by LRRC15.

## LRRC15 accumulates cell-attached viruses on the membrane and does not compete with ACE2

As SARS-CoV-2 entry is primarily dependent on ACE2, we assessed whether LRRC15 alters protein expression of ACE2. The level of ACE2 surface expression was unaltered or marginally decreased by sgRNA-mediated gene induction in HeLa-ACE2 cells and was slightly increased in LRRC15-induced Huh7.5 cells (**Figs 4A and S4A**). Although surface ACE2 expression was slightly different, they all showed the decreased entry of SARS-CoV-2 pseudotyped virus and authentic virus, indicating that the inhibitory effect of LRRC15 does not require the regulation of surface ACE2 levels.

Interestingly, we found that spike-coated viruses were sequestered on the cellular surface of LRRC15-expressing cells. In the attachment assay, we measured viral attachment to cells by incubating spike-pseudotyped viruses and cells on ice, allowing attachment on the cell membrane and preventing internalization of viruses. As expected, SARS-CoV-2 spike-pseudotyped viruses bound to ACE2-expressing cells and the binding was not altered by inducing a control

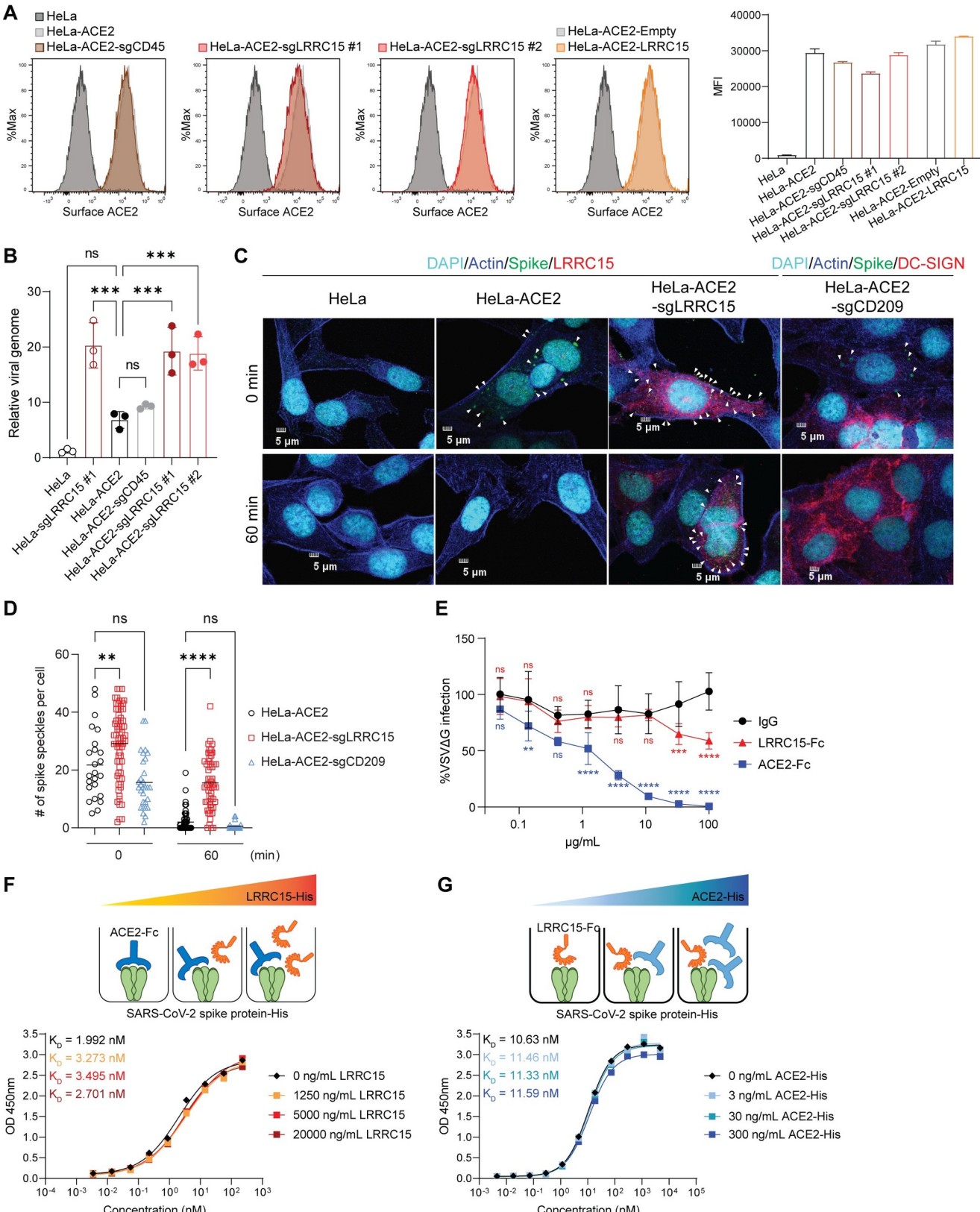

**Fig 4. LRRC15 enhances SARS-CoV-2 attachment to the surface of ACE2-expressing cells.** (**A**) HeLa-ACE2 cells were transduced with indicated activating sgRNAs or a LRRC15-expressing vector. Cell surface expression of ACE2 was measured by flow cytometry and calculated as MFI (*n* = 3). (**B**)

HeLa or HeLa-ACE2 cells transduced with indicated activating sgRNAs were incubated with VSVΔG-S-SARS2 for 1 h on ice and washed 3 times with cold cell culture media. Viral genome copies were quantified by RT-qPCR and normalized to HeLa-ACE2 cells ($n = 3$). (**C**) Representative images of immunofluorescence staining of SARS-CoV-2 spike (green), LRRC15 (red), Actin (blue), and DAPI (cyan). Cells were inoculated with VSVΔG-S-SARS2 for 1 h on ice and incubated at 37°C for 1 h to allow internalization, followed by staining. The white arrowheads indicate spikes. The scale bar indicates 5 μm. (**D**) Quantification was performed by calculating the number of spikes on cells from multiple images per sample. (**E**) VSVΔG-S-SARS2 were incubated with ACE2-Fc, LRRC15-Fc, or IgG control for 1 h, prior to inoculating HeLa-ACE2 cells. Viral infectivity was quantified by measuring GFP signal at 20 hpi by flow cytometry and normalized to no antibody control ($n = 6$). Statistical significance was determined compared to IgG control at each dilution. (**F, G**) Competition assays between ACE2 and LRRC15 for immobilized His-tagged SARS-CoV-2 spike protein. Premixture of His-tagged protein at 4 different concentrations with a dilution series of Fc-tagged protein was added, and antihuman HRP determined the amount of Fc-tagged proteins remaining in the presence of competitor through a colorimetric readout. A combination of LRRC15-His and ACE2-Fc (**F**) or ACE2-His and LRRC15-Fc (**G**) was used. Data represent means ± SD (**B**, **D**, **E**). Data were analyzed by one-way ANOVA (**B**, **D**) or two-way ANOVA (**E**) with Dunnett multiple comparisons test. ns, not significant; $^*p < 0.05$; $^{**}p < 0.01$; $^{***}p < 0.001$; $^{****}p < 0.0001$. For underlying data, see **S3 Data**. ACE2, angiotensin-converting enzyme 2; GFP, green fluorescent protein; hpi, hours postinfection; HRP, horseradish peroxidase; IgG, immunoglobulin G; LRRC15, leucin-rich repeat-containing 15; MFI, mean fluorescence intensity; RT-qPCR, quantitative reverse transcription PCR; SARS-CoV-2, Severe Acute Respiratory Syndrome Coronavirus 2; sgRNA, single guide RNA.

gene, CD45. Importantly, viruses were highly accumulated on LRRC15-induced cells, both in the absence and presence of ACE2 (i.e., approximately 3-fold increases in viral copies) compared to their controls (**Fig 4B**). The enhanced virus binding by LRRC15 was observed independent of expression of ACE2. Immunofluorescence staining of spike proteins confirmed the enhanced binding of pseudoviruses to LRRC15- but not to DC-SIGN-induced cells (**Figs 4C, 4D and S4B**). While few spikes were detected in HeLa-ACE2 cells after 1 h of internalization, significant number of spikes were detected in LRRC15-induced cells, implying that LRRC15 may inhibit the internalization of the virus (**Fig 4D**). Preincubation of soluble LRRC15 protein with spike-pseudotyped viruses partially blocked viral entry in HeLa-ACE2 cells at high concentrations, while preincubation with soluble ACE2 completely blocked viral entry (**Fig 4E**). The difference in blocking efficacy is likely due to the differing spike binding affinities of LRRC15 and ACE2. In summary, these results suggest that LRRC15 inhibits SARS-CoV-2 entry potentially by restricting the internalization of virions into the cell through binding to the spike protein.

To test whether LRRC15 directly binds ACE2, we utilized His-tagged-LRRC15, SARS-CoV-2 spike protein, and MERS-CoV spike protein and assessed their interactions with Fc-tagged ACE2. Recombinant ACE2 did not show detectable binding to recombinant LRRC15 protein or the MERS-CoV spike whereas binding to SARS-CoV-2 spike was confirmed with high affinity (**S4C Fig**) [39]. Since both ACE2 and LRRC15 bind to the RBD of SARS-CoV-2 spike, we investigated whether LRRC15 competes with ACE2 for binding on the spike protein. The interaction between ACE2 and SARS-CoV-2 spike was measured in the presence of recombinant LRRC15 protein. Even at high concentrations, LRRC15 did not affect the spike-ACE2 binding (**Fig 4F**). Conversely, spike-LRRC15 interaction was not affected by excess ACE2, demonstrating LRRC15 and ACE2 do not share the same binding epitope within the RBD (**Fig 4G**). We confirmed that SARS-CoV-2 spike S1-Fc binding to Hela-ACE2 cells was not altered by gene induction or ectopic expression of LRRC15 (**S4D Fig**). Taken together, ACE2-mediated viral entry of SARS-CoV-2 is suppressed by LRRC15 on the cell membrane through its direct binding to the RBD without competition between LRRC15 and ACE2.

## LRRC15 is expressed in distinct cell types in human lung and is associated with pathological fibroblasts

To better understand the function of LRRC15 in a physiologic context, we first explored its expression in human lung samples unaffected by SARS-CoV-2. Two independent single-cell RNA-sequencing (scRNA-seq) datasets of non-COVID-19 human lungs [40,41] revealed that *LRRC15* was predominantly expressed in a subset of fibroblasts and lymphatic endothelial cells

(**S5A and S5B Fig**). For instance, in the Tissue Stability Cell Atlas dataset, a significant proportion of fibroblasts and lymphatic endothelial cells expressed *LRRC15* (**S5B Fig**). Of note, the cell types that expressed *LRRC15* did not coexpress *ACE2*.

Having defined fibroblasts and lymphatic endothelial cells as the main cell types in the lung that express *LRRC15*, we sought to explore if any clinical features were associated with *LRRC15* expression in the lung. Utilizing the large cohort of human lung RNA-seq samples from the Genotype-Tissue Expression (GTEx) project [42], we constructed a multivariable regression model between *LRRC15* expression and various clinical factors as predictors. Specifically, we included age, sex, diabetes (type 1 or type 2), hypertension, body mass index (BMI), smoking, and ventilator status at time of death as independent variables in the model. We observed that *LRRC15* expression was not significantly associated with age, sex, hypertension, BMI, smoking, or type 1 diabetes (**S5C Fig**). Strikingly, *LRRC15* expression was significantly decreased in patients that were on a ventilator prior to death (**S5D Fig**). We note that the causality of this association cannot be ascertained due to the retrospective nature of the dataset: It is unclear whether ventilator usage leads to lower *LRRC15* expression, or whether patients with conditions that subsequently require mechanical ventilation have baseline alterations in lung physiology associated with decreased *LRRC15* expression.

In order to gain further insight on *LRRC15* expression in the lung, we calculated the correlation between *LRRC15* and all other genes in the GTEx lung dataset. We observed that genes such as *SOX4*, *FRMD6*, *FAP*, *ENAH*, *PRRX1*, *CD200*, and *VCAM1* were positively correlated with *LRRC15* (**S5E Fig**). In contrast, *ACE2* was negatively correlated with *LRRC15* (**S5F Fig**), which is consistent with their distinct cell type–specific expression patterns (**S5A and S5B Fig**). We subsequently mapped these highly correlated genes to specific lung cell types, finding that fibroblasts and lymphatic endothelial cells also expressed several of the genes that were positively correlated with *LRRC15* (**S5G Fig**). We further observed that *MAOA*, which showed the strongest negative correlation with *LRRC15* in the bulk lung RNA-seq cohort, was mostly expressed in alveolar type 2 cells (**S5E Fig**).

To explore the clinical relevance of LRRC15 to COVID-19 pathophysiology, we analyzed 2 independent scRNA-seq datasets of lungs from deceased COVID-19 patients [43,44]. Consistent with our analyses of non-COVID-19 lungs, we found that fibroblasts and lymphatic endothelial cells had the highest levels of *LRRC15* expression (**Fig 5A and 5B**). Interestingly, we observed that *LRRC15* expression was particularly enriched in the pathological fibroblast subpopulation. This recently identified fibroblast subset (defined by high *CTHRC1* expression) has been implicated as a key contributor to idiopathic pulmonary fibrosis [45] and may also drive lung fibrosis in COVID-19 patients [43]. Indeed, the relative proportion of pathological fibroblasts and intermediate pathological fibroblasts was significantly increased in COVID-19 patients compared to controls (**Fig 5C**). Further supporting the association between *LRRC15* and disease-associated fibroblast cell states, there was a progressive gradient of *LRRC15* expression from alveolar fibroblasts (0.26%) to intermediate pathological fibroblasts (2.61%) and, finally, to pathological fibroblasts (4.52%).

Given that our experiments pointed to LRRC15 as an antiviral restriction factor, we sought to specifically investigate whether *LRRC15* is associated with viral burden and disease progression in COVID-19 patients. We analyzed a bulk RNA-seq dataset of lungs from COVID-19 patients [46] with high or low SARS-CoV-2 viral burden at the time of autopsy. While patients with high versus low viral burden might correspond to 2 distinct disease phenotypes, patients with high viral RNA load had experienced shorter duration of illness before death, suggesting that these patients had failed to control the virus and died during the acute phase of infection [46]. On the other hand, patients with low viral RNA load had longer duration of illness before death, consistent with a scenario in which they had successfully controlled the virus but

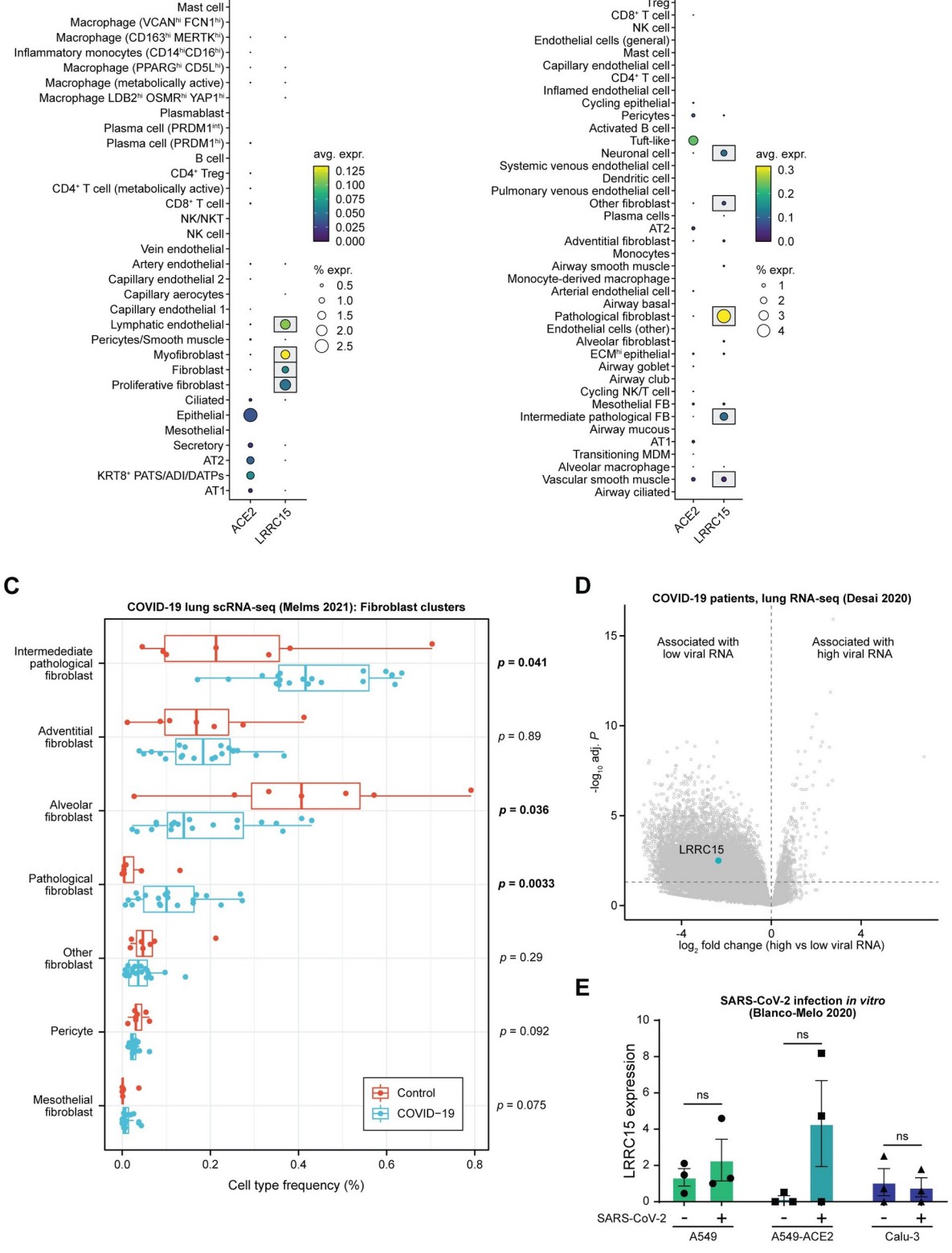

**Fig 5. LRRC15 expression is enriched in the fibroblasts of COVID-19 patients and associated with reduced SARS-CoV-2 viral burden.** (**A**, **B**) Cell type–specific expression of *ACE2* and *LRRC15*, assessed by scRNA-seq of lungs from deceased COVID-19 patients (**A**)

Delorey and colleagues [44] or (**B**) Melms and colleagues [43]. (**C**) Tukey boxplots (IQR boxes with $1.5 \times$ IQR whiskers) of the relative frequencies of fibroblast subtypes among total fibroblasts, comparing COVID-19 patients (blue) to non-COVID-19 controls (red). Statistical significance was assessed by two-tailed unpaired Mann–Whitney test. (**D**) Volcano plot of differentially expressed genes in the lungs of deceased COVID-19 patients, comparing samples with high vs. low SARS-CoV-2 RNA levels at the time of death [46]. Genes with positive $\log_2$ fold changes are associated with high viral burden, while genes with negative $\log_2$ fold changes are associated with low viral burden. (**E**) *LRRC15* expression in lung cell lines (A549, A549-ACE2, Calu-3), comparing mock controls vs. SARS-CoV-2-infected samples. Data represent means ± SEM. Statistical significance was assessed by two-tailed unpaired Welch's *t* test. For underlying data, see **S2 Data**. ACE2, angiotensin-converting enzyme 2; COVID-19, Coronavirus Disease 2019; IQR, interquartile range; LRRC15, leucin-rich repeat-containing 15; SARS-CoV-2, Severe Acute Respiratory Syndrome Coronavirus 2; scRNA-seq, single-cell RNA-sequencing.

subsequently died from sequalae of the infection. With this framework in mind, we found that expression of *LRRC15* was significantly higher in patients with low SARS-CoV-2 viral burden (**Fig 5D**). Though the causality underlying this relationship is unclear, it is supportive of the antiviral function of LRRC15 that we have identified here. Of note, SARS-CoV-2 infection of lung epithelial cell lines in vitro did not lead to significant changes in *LRRC15* expression (**Fig 5E**), which is consistent with our prior analyses pointing to a fibroblast-specific role of *LRRC15* in COVID-19. Collectively, our analyses point to a model in which SARS-CoV-2 infection induces the emergence of a pathological fibroblast state with increased LRRC15 expression.

## LRRC15 in ACE2-negative cells inhibits SARS-CoV-2 infection *in trans*

Since the scRNA-seq analysis revealed that *LRRC15* is not coexpressed with *ACE2* within the same cell types in the lung, we hypothesized that LRRC15 inhibits SARS-CoV-2 entry into ACE2-positive cells *in trans*. To test this hypothesis, HeLa-ACE2 cells were cocultured with either HeLa-control or HeLa-sgLRRC15 cells (i.e., ACE2-negative cells) and were subsequently infected with spike-pseudotyped viruses (**Fig 6A**). The GFP reporter signal was only detected in the ACE2+ cells, confirming LRRC15 alone does not permit viral entry (**S6A Fig**). Compared to the control HeLa cells, coculturing with HeLa-sgLRRC15 cells resulted in a significant reduction of viral entry in ACE2+ cells when cocultured at a 1:4 ratio (**Fig 6B**). At 2 different titers of viral infection, the similar pattern of reduction was observed and coculturing with DC-SIGN-expressing cells did not induce the reduction, confirming the specificity of inhibitory function of LRRC15. The same trend of *trans*-inhibition was confirmed by the spike-pseudotyped viral infection with the Delta (B.1.617.2) variant (**Fig 6C**). Coculture of HeLa-ACE2 cells and HeLa-sgLRRC15 cells at a 1:1 ratio exhibited significant restriction activities, although the magnitude of suppression was slightly weaker as expected (**S6B and S6C Fig**). In lungs from COVID-19 patients [43], the scRNA-seq data revealed that *LRRC15*-expressing fibroblasts and endothelial cells significantly outnumbered *ACE2*-expressing epithelial cells (**S5H Fig**). Specifically, 70% of the samples (14 of 20) showed greater frequencies of *LRRC15*-expressing fibroblasts and endothelial cells than *ACE2*-expressing epithelial cells, supporting the relevance of the cell mixing ratios used in **Fig 6B and 6C**.

Immunofluorescence staining with the coculture model further confirmed the spike binding on LRRC15+ cells and strong colocalization of LRRC15 and spike on these cells (**Fig 6D**). Spike-pseudotyped virus-infected cells exhibited speckle-like staining patterns for spike. The coculture condition with HeLa-sgLRRC15 showed that a significant proportion of spike speckles were detected on LRRC15+ cells. Most interestingly, spike speckles on the coculture condition with HeLa-sgLRRC15 cells retained overtime without further entry progress, whereas the spike speckles rapidly disappeared in cells within 60 min in the coculture condition with wild-type HeLa cells (**Fig 6E**). These results indicate that spike binding on LRRC15+ cells is not functional binding for entry, but for noninfectious sequestration of virions. Collectively, these

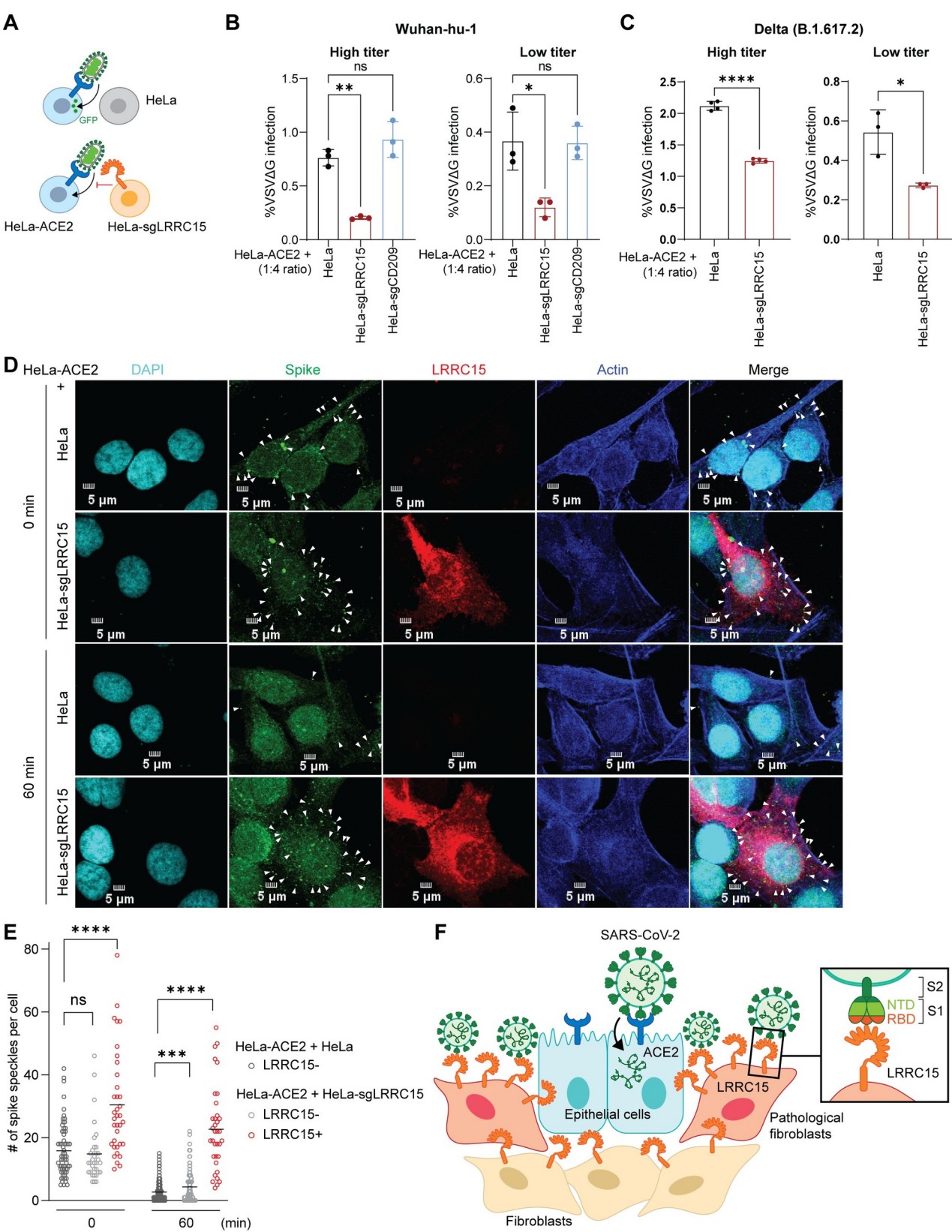

**Fig 6. LRRC15 inhibits ACE2-mediated SARS-CoV-2 entry *in trans*.** (**A**) Schematic of *trans*-inhibition assay with ACE2+ and LRRC15+ cells. (**B**) HeLa-ACE2 cells were cocultured with HeLa, HeLa-sgLRRC15, or HeLa-sgCD209 cells at 1:4 ratio and infected with VSV pseudovirus–harboring spike of SARS-CoV-2 Wuhan-hu-1 strain at high or low titer. GFP signal was measured at 20 hpi by flow cytometry (*n* = 3). Representative of 3 independent experiments are shown. (**C**) *Trans*-inhibition assay was performed as in (**B**) with VSV pseudovirus–harboring spike of SARS-CoV-2 Delta variant (B.1.617.2 strain) (*n* = 4). Representative of 3 independent experiments are shown. (**D**) Representative images of immunofluorescence staining of SARS-CoV-2 spike (green), LRRC15 (red), Actin (blue), and DAPI (cyan). HeLa-ACE2 cells were cocultured with HeLa or HeLa-sgLRRC15 cells, inoculated with VSVΔG-S-SARS2 for 1 h on ice and incubated at 37˚C to allow internalization. Cells were harvested after 1 h and subjected to staining. The white arrowheads indicate spike. The scale bar indicates 5 μm. (**E**) Quantification was performed by calculating the number of spikes on LRRC15− or LRRC15+ cells from multiple images per sample. (**F**) Proposed working model of the LRRC15-mediated inhibition of SARS-CoV-2 entry. Data represent means ± SD (**B**, **C**, **E**). Data were analyzed by one-way ANOVA with Dunnett multiple comparisons test (**B**, **E**) or unpaired two-tailed *t* test (**C**). ns, not significant; $^*p < 0.05$; $^{**}p < 0.01$; $^{***}p < 0.001$; $^{****}p < 0.0001$. For underlying data, see **S3 Data**. ACE2, angiotensin-converting enzyme 2; GFP, green fluorescent protein; hpi, hours postinfection; LRRC15, leucin-rich repeat-containing 15; SARS-CoV-2, Severe Acute Respiratory Syndrome Coronavirus 2; VSV, vesicular stomatitis virus.

results highlight the *trans*-inhibitory function of LRRC15 proven using a coculture model, suggesting that expression of LRRC15 in SARS-CoV-2-nonpermissive fibroblasts protects permissive cells against viral infection in the lung (**Fig 6F**).

## Discussion

In this study, we employed a *surface*ome CRISPRa library to identify cellular factors for SARS-CoV-2 by staining cells with a recombinant spike protein. The *surface*ome screening revealed 2 distinct hits: ACE2, the bona fide entry receptor, and LRRC15, a novel inhibitory attachment factor. Binding assays using recombinant proteins in cells and in cell-free models showed that LRRC15 directly interacts with the RBD of spike with a moderate affinity ($K_D$ = approximately 43 to 148 nM, depending on domain and variant). Although ACE2 also interacts with the spike via the RBD, the interaction of LRRC15-RBD neither competes nor stabilizes the interaction of ACE2-RBD. Further studies will be necessary to determine the interface of LRRC15-RBD at a higher resolution.

The inhibition of viral entry by LRRC15 represents a direct effector by interacting with the spike protein itself. The leucin-rich repeat (LRR) domain is functionally linked with sensing of pathogen-associated molecular patterns (PAMPs) in a number of cases [47]. The LRR domain proteins are highly conserved throughout evolution including in plants, providing a prominent protection against pathogens [48,49]. Given the role of LRR domains in pattern recognition, humans might develop a pattern recognition protein for certain types of coronaviruses, in this case SARS-CoV-1 and SARS-CoV-2. The robust and specific inhibitory effect of LRRC15 on multiple variants of SARS-CoV-2 and SARS-CoV-1 (but not MERS-CoV) suggests an arms race between humans and coronaviruses. While this manuscript was in preparation, LRRC15 was also identified as a SARS-CoV-2 spike-binding factor in 2 preprints [50,51]. This demonstrates the LRRC15-spike interaction is robust and detectable in multiple cell line models. Albeit the mechanism of viral restriction is different, several interferon-stimulated genes (ISGs) such as LY6E, CH25H, and IFITMs are known to restrict coronavirus entry by interfering with spike protein-mediated membrane fusion [52–54] or by interfering with endosome-mediated processes [55,56]. Unlike these ISGs, LRRC15-mediated inhibition of viral entry is directly mediated by interaction with the spike, which resembles PAMP receptors. LRRC15 is not induced by interferons (**S6D Fig**). As LRRC15 redistributes adenovirus receptor (CAR) affecting the delivery of adenovirus to cells, it seems that LRRC15 may act as an antiviral factor via different modes [57]. It remains to be elucidated whether LRRC15 requires intracellular signaling through its cytosolic domain or requires other cellular proteins for the inhibition. As we did not see an effective restriction activity of LRRC15 on cells overexpressing TMPRSS2 (**S3I Fig**), it is possible that LRRC15 exhibits differential restriction activities depending on entry pathways of SARS-CoV-2. Further investigation for the differential sensitivity will be

necessary to determine the dependence of enzymatic entry cofactors including TMPRSS2, cathepsins, or metalloproteinases in different cell types with their endogenous expression.

Most strikingly, LRRC15 inhibited spike-mediated viral entry not only in the same cells, but also in neighboring cells *in trans*, providing a unique concept of viral entry inhibition by an inhibitory attachment factor. The coculture of ACE2+LRRC15− cells and ACE2-LRRC15 + cells exhibited a significant suppression of viral infection with spike proteins of 2 different SARS-CoV-2 variants. It suggests that the antiviral effect of LRRC15 might be functional in a broader physiological context, which does not require cell-intrinsic restriction in SARS-CoV-2 susceptible cells in the lung, although such inhibitory function may vary, depending on relative distribution and expression of LRRC15 and ACE2 in vivo. During SARS-CoV-2 infection in the lung of human patients, LRRC15 may promote viral control by functioning as an entry inhibitor, likely in a non-cell-autonomous manner given that *ACE2* and *LRRC15* are expressed in distinct (but potentially neighboring) cell types. Thus, while the emergence of a *LRRC15*[+] pathological fibroblast population may ultimately drive the fibrotic changes observed in patients with COVID-19, these fibroblasts may initially play a protective role by contributing to viral clearance during the acute phase of infection through their expression of LRRC15 on the cell surface, subsequently paving the way for the transition toward tissue repair and remodeling. Importantly, while this article was under review, a multiomics study identified LRRC15 as a strong predictor of COVID-19 severity in human patients in a preprint [58].

In conclusion, this work reveals that LRRC15 is an attachment factor for SARS-CoV-2 spike and represents a pattern recognition–like inhibition of viral entry by directly interacting with the spike protein. This study provides insight into therapeutic development and a better understanding of COVID-19.

## Materials and methods

### Cell culture

HEK293T (#CRL-3216), HeLa (#CCL-2), A375 (#CRL-1619), and Huh7.5 (#CVCL-7927) cells were purchased from ATCC. HEK293T-ACE2/TMPRSS2 cells were obtained through BEI Resources, NIAID, NIH (#NR-55293). Cells were cultured in Dulbecco's Modified Eagle Medium (DMEM, Gibco, #11995–081) supplemented with 10% fetal bovine serum (FBS) and 2.5% HEPES (Gibco, #15630–080) and detached using 0.05% trypsin-EDTA with phenol red (Gibco, #25300–120). After transfection, viral production media (DMEM with 10% FBS, 2.5% HEPES, and 1% bovine serum albumin) was used for lentiviral production. For A375 cells, 5 μg/mL blasticidin (Gibco, #A1113903) and 1 μg/mL puromycin (Gibco, #A1113803) were added as appropriate. For HeLa cells, 5 μg/mL blasticidin, 0.7 μg/mL puromycin, 200 or 400 μg/mL hygromycin (Gibco, #0-687-010), and 100 μg/mL zeocin (Thermo, #R25001) were added as appropriate. Huh7.5 cells were treated with 5 μg/mL blasticidin and 4 μg/mL puromycin as appropriate.

### Generation of genetically modified cell lines

Individual sgRNAs (sgLRRC15 #1: GACATGCAGGCACTGCACTG; sgLRRC15 #2: AGTGT-CAGCCCGGGACATGC; sgACE2: GTTACATATCTGTCCTCTCC; sgCD209: AAAGCAT-CAGAGCATGAGAT) targeting the candidate genes were cloned into linearized pXPR_502 (Addgene, #96923) for CRISPRa. Media was replaced with viral production media 12 h after transfection. The supernatant was collected, spun at 4,347 × *g*, and filtered using a 0.45-μm filter (Millipore) 36 h after transfection. A375-dCas, HeLa-dCas, or Huh7.5-dCas cells were generated by transducing with pLenti-dCas9-VP64-Blast (Addgene, #61425). After 7 d of blasticidin selection, cells were transduced with 1 mL of harvested lentiviral stock and spun at

$1,200 \times g$ for 90 min at 35˚C. Cells were given 2 mL of fresh media and were incubated for 6 or 18 h after spin transduction. Puromycin was used to select for successfully transduced cells.

Stable ACE2 expressing HeLa cells (HeLa-ACE2) were generated by transducing HeLa-dCas cells with pLENTI_hACE2_HygR (Addgene, #155296) followed selection with hygromycin. To generate a clonal cell line, transduced cells were plated in 96-well plates at single cell dilutions. After propagating the cells, each clone was screened for surface ACE2 expression by flow cytometry, and a clone with the highest ACE2 expression was selected and used for subsequent experiments.

For ectopic expression of LRRC15, a lentiviral vector pCDH-MSCV-T2A-Puro (System Biosciences, #CD522A-1) was modified to enable zeocin selection (pCDH-MSCV-T2A-Zeo) or to contain an EF1alpha promoter, beta-globin intron, and internal ribosome entry site (pCDH-EF1a-intron-IRES-Puro). A codon-optimized LRRC15 ORF with a C-terminal 3xFLAG tag was cloned into pCDH-MSCV-T2A-Zeo and used to transduce HeLa-ACE2 and pCDH-EF1a-intron-IRES-Puro to transduce HEK293T-ACE2/TMPRSS2, followed by zeocin or puromycin selection.

## CRISPR activation screen for SARS-CoV-2 spike binding

A list of 6,011 surface proteins was obtained by integrating 4 datasets for plasma membrane proteins [59–62]. Four sgRNA sequences targeting each gene were picked from Calabrese genome-wide CRISPRa library [63], and 1,000 nontargeting control sgRNAs were included. The sgRNAs were cloned into pXPR_502 (Addgene, #96923) with assistance from the Genome Engineering and iPSC Center (GEiC) at Washington University in Saint Louis. To make $2.4 \times 10^7$ transduced cells, $7.8 \times 10^7$ A375-dCas cells were transduced with the CRISPRa library at approximately 0.3 MOI, which is sufficient for the integration of each sgRNA into approximately 500 cells. At 2 d posttransduction, puromycin was added and cells were selected for over a week.

For SARS-CoV-2 spike S1-Fc binding screen, $5 \times 10^7$ cells per sample were washed with FACS buffer (1× PBS supplemented with 2% FBS and 1 mM EDTA) and incubated with 50 μg/mL SARS-CoV-2 spike S1 subunit-Fc fusion protein (R&D Systems, #10623-CV-100) or human IgG1 isotype control (BioXCell, #BE0297) for 30 min at 4˚C. After washing 2 times with FACS buffer, cells were stained with PE-conjugated antihuman IgG antibody (Southern Biotech, #9040–09) for 30 min at 4˚C. Then, the cells were washed 2 times with FACS buffer, fixed with 4% formaldehyde, and subjected to sorting using FACSAriaIII (BD Biosciences). The top approximately 3% (fluorescence intensity) of the PE-positive cells were isolated. As a control, a same number of cells were stained with BV421 anti-hCD45 antibody (BioLegend, #368522) and the top 3% of the BV421-positive cells were sorted. Genomic DNA (gDNA) was extracted from the isolated cells and unsorted cells ("Input") with QIAamp DNA Maxi kit (Qiagen, #51104).

## CRISPR screen sequencing and analysis

For Illumina sequencing, gDNA was used for PCR to amplify the integrated sgRNA sequences. PCR was performed in 96-well plates and each well containing up to 10 μg of gDNA in a total of 100 μL reaction mixture consisting of Titanium Taq DNA polymerase buffer and enzyme (Takara, #639209), deoxynucleoside triphosphate, dimethylsulfoxide (5%), P5 stagger primer mix (0.5 μM), and uniquely barcoded P7 primer (0.5 μM). Samples were amplified with following PCR cycles: an initial 5 min at 95˚C; followed by 28 or 30 cycles of 95˚C for 30 s, 59˚C for 30 s, and 72˚C for 20 s; followed by a final 10 min at 72˚C. PCR products from 2 wells per sample were pooled and purified with AMPure XP beads according to the manufacturer's protocol

(Beckman Coulter, #MSPP-A63880). Samples were sequenced on a NextSeq550 sequencer (Illumina). After demultiplexing according to the barcode sequences, reads were mapped to a reference file of sgRNAs in the surface CRISPRa library using LibraryAligner (https://gitlab. com/buchserlab/library-aligner). We calculated the log-fold change of sgRNAs in each sample relative to the unsorted cells and calculated the hypergeometric distribution to determine *p*-values.

## Flow cytometry

For SARS-CoV-2 spike subunit binding assay, cells were washed once with HBSS containing 2% FBS and incubated with 50 μg/mL S1-Fc, 200 μg/mL RBD-Fc (Sino Biological, #40592-V08H) or NTD-Fc (Sino Biological, #40591-V49H) for 30 min at 4°C, followed by washing 2 times with HBSS with 2% FBS. Cells were incubated with PE antihuman IgG for another 30 min at 4°C, washed 2 times, and fixed with 4% formaldehyde for 15 min. Cells were washed once, resuspended in HBSS with 2% FBS, and analyzed by flow cytometry using FACSCelesta (BD Biosciences). A full-length spike protein including a T4 foldon trimerization domain (BEI Resources, NR-53937) was used at approximately 100 to 1,000 μg/mL, followed by staining with anti-6xHis (Thermo Scientific, #MA1-135) and FITC anti-mIgG1 (BD, #553443). To measure the surface expression of ACE2 and LRRC15, cells were washed with FACS buffer (1× PBS supplemented with 2% FBS and 1 mM EDTA) and stained with goat anti-ACE2 (R&D Systems, #AF933) at a 1:50 dilution or rabbit anti-LRRC15 (Abcam, #ab150376) at a 1:100 dilution for 30 min at 4°C. Then, the cells were washed 2 times and resuspended in FACS buffer containing the secondary antibodies at a 1:1,000 dilution: AF647-labeled donkey anti-goat IgG (Invitrogen, #A32849) or AF488-labeled goat anti-rabbit IgG (Invitrogen, #A32731). After 30 min incubation at 4°C, the cells were washed 2 times, fixed with 4% formaldehyde for 15 min, and washed and resuspended in FACS buffer before analyzing by flow cytometry using FACSCelesta (BD Biosciences) or Cytek Aurora spectral analyzer (Cytek Biosciences). Data were analyzed with FlowJo software, and cells were gated as represented in **S7 Fig**.

## ELISA binding assay

To investigate the binding of LRRC15 to SARS-CoV2 spike proteins, ELISA assays were performed on immobilized spike protein-Fc. To this aim, 96-well Immulon 2HB flat bottom plates (Thermo) were coated with 2 μg/mL spike proteins with C-terminal Histidine (BEI Resources, NR-55438, NR-55311, NR-55310, NR-52724, NR-53769, and NR-55307, NR-55614, and NR-53589) at 4°C overnight, followed by 1-h blocking buffer containing 1× HBSS (Gibco, #14025–092) and 2% FBS (VWR, #104B16). The plates were then incubated with either ACE2-Fc (Sino Biological, #10108-H02H, starting concentration, 10 μg/mL), LRRC15-Fc (Sino Biological, #15786-H02H, 100 μg/mL) or human IgG1 isotype (BioXCell, #BE0297) (as negative control) serially diluted 4-fold for 2 h. The plates were then incubated with goat anti-human IgG Fc secondary antibody, HRP (Thermo, #A18817) at a 1:3,000 dilution for 1 h at room temperature. Next, TMB substrate (Thermo Scientific, #ENN301) was added to the plates and then quenched with stop solution (Thermo Scientific, #PIN600). Absorbance at 450 nm were recorded with a BioTek synergy HT microplate reader. Three washes were performed between every incubation using 1× HBSS with 0.05% Tween-20. GraphPad Prism 9 software was used to perform nonlinear regression curve-fitting analyses of binding data to estimate dissociation constants ($K_D$).

To determine which SARS-CoV-2 spike protein region contributes to the binding of LRRC15, the ELISA was performed by coating 96-well plates with 2 μg/mL LRRC15-His

(AcroBio, #LR5-H52H3) in coating buffer (BD, #51-2713KC) overnight at 4˚C. Following blocking with 2% FBS, the 4-fold serially diluted SARS-CoV-2 spike RBD-Fc recombinant protein (Sino Biological, #40592-V02H) and SARS-CoV-2 spike S1 NTD-Fc (Sino Biological, #40591-V41H) were added and incubated for 2 h at room temperature. After washing, the HRP-labeled goat antihuman IgG Fc secondary antibody was added and incubated for 1 h. Subsequently, TMB substrate was added, and the enzymatic reaction was stopped by adding stop solution. The signal was read at 450 nm.

To compare the binding affinity of LRRC15 and ACE2 with other coronaviruses, the 96-well plates were coated overnight at 4˚C with 2 μg/mL LRRC15-His or ACE2-His (Sino Biological, #10108-H08H). This was followed by blocking with 2% FBS for 1 h at room temperature. Then, either rabbit Fc-tagged-MERS-CoV spike/RBD protein fragment (Sino Biological, #40071-V31B1) or SARS-CoV spike/RBD protein fragment (Sino Biological, #40150-V31B2) were 4-fold serially diluted (starting concentration, 16 μg/mL) and added on the plate for 2-h incubation. Later, goat anti-rabbit IgG-HRP conjugate (Southern Biotech, #4030–05) diluted (1:3,000) in HBSS was used to detect the bound MERS or SARS-CoV RBD fragment. The reaction of HRP with TMB developed a colorimetric signal. The absorbance value was read at 450 nm after adding stop solution.

To determine whether LRRC15 binds to ACE2, 96-well plates were coated with either 2 μg/mL LRRC15-His, spike glycoprotein from SARS-CoV-2 with C-terminal histidine (BEI Resources, NR-53589), or spike glycoprotein from MERS-CoV, England 1 with C-terminal histidine (BEI Resources, NR-53591) and incubated overnight at 4˚C. After blocking at room temperature for 1 h with 2% FBS in HBSS, the ELISA plates were washed, and 4-fold serially diluted ACE2-Fc (starting from 160 μg/mL) were added for a 2-h incubation. HRP-conjugated goat antihuman IgG Fc secondary antibody was used to detect the bound ACE2.

## ELISA competition assay

The ACE2-Fc were serially diluted 4-fold starting from 40 μg/mL. Each dilution was mixed with different concentrations of LRRC15-His (0, 1,250, 5,000, and 20,000 ng/mL). These sample series were transferred to the plate that coated with spike glycoprotein from SARS-CoV-2, Wuhan-Hu-1 with C-terminal histidine (BEI Resources, NR-53947) at 4˚C, overnight. After 2-h incubation, the wells were treated with goat antihuman IgG Fc secondary antibody, HRP at a 1:3,000 dilution for 1 h at room temperature. Chromogenic development was generated with TMB substrate and quenched with stop solution. Optical density was measured in a Bio-Tek synergy HT microplate reader.

For spike protein competition, ACE2-His (Sino Biological, #10108-H08H) was diluted to 300 ng/mL, 30 ng/mL, 3 ng/mL. The ACE2-His was added to the plate with serially diluted LRRC15-Fc, allowing ACE2-His to compete with LRRC15-Fc binding to spike protein (BEI Resources, NR-53947) immobilized on the plate. After 2-h incubation, the plate was then washed, incubated with goat antihuman IgG Fc secondary antibody and the signal detected as described above.

## Pseudovirus production

VSV-dG pseudoviral particles were produced as previously described [34]. Briefly, $8 \times 10^6$ HEK293T cells were plated in 10-cm tissue culture dishes and transfected using Lipofectamine2000 (Invitrogen) with plasmids encoding different CoV spike proteins or VSV-G protein. Expression vectors for SARS-CoV-2 Wuhan-Hu-1 (Addgene, #149539), SARS-CoV-2 B.1.167.2 (Addgene, #172320), SARS-CoV-2 B.1.1.7 (Addgene, #170451), SARS-CoV-2 B.1.351 (Addgene, #170449), SARS-CoV-2 P.1 (Addgene, #170450), SARS-CoV-1 (Addgene,

#170447), MERS-CoV (Addgene, #170448), and VSV-G (Addgene, #12259) were used. At 24 h posttransfection, cells were incubated with replication restricted rVSVΔG*G-GFP virus (Kerafast, #EH1019-PM) at approximately 5 MOI for 1 h at 37°C, 5% $CO_2$ and the media was replaced with complete media. Anti-VSV-G (Sigma, #MABF2337) was added at final concentration of 1 μg/mL to neutralize residual rVSVΔG*G. At approximately 24 h postinoculation, viral supernatant was harvested, cell debris were removed by centrifuging for 10 min at 1,320 × $g$, and stored at −80°C in small aliquots.

## Pseudovirus entry assay

Cells were plated at $1 \times 10^5$ cells per well in 24-well plates or $2 \times 10^4$ cells per well in 96-well plates. The following day, media was removed from the cells and 150 μL or 50 μL of pseudotyped VSV were added. After incubating 1 h at 37°C, 5% $CO_2$, virus-containing media was removed and the cells were incubated in complete media for approximately 20 to 24 h at 37°C, 5% $CO_2$. Cells were washed once and resuspended with FACS buffer and GFP-positive cells were measured by flow cytometry using FACSCelesta (BD Biosciences) and single cells were analyzed following the gating strategy shown in S7 Fig. When indicated, cells were pretreated with 20 μM Camostat mesylate (Sigma) for 1 h then subjected to VSV infection. For neutralization assay, SARS-CoV-2 spike-pseudotyped VSV was preincubated with serial 3-fold dilutions of ACE2-Fc, LRRC15-Fc, or human IgG control for 1 h at 37°C, 5% $CO_2$ before adding to the cells.

## SARS-CoV-2 mNeonGreen virus infection assay

Procedure to infect the different LRRC15 cell lines with SARS-CoV-2 mNeonGreen WA01 was followed as previously described [64]. Briefly, cells were seeded at $3 \times 10^6$ per well in a 384-well plate a day prior infection. SARS-CoV-2 mNG WA01 [36] was used at an MOI of 1.0 to infect for 24 h. Infection efficiency was measured by expression of mNeonGreen using high content imaging (Cytation 5, BioTek) configured with both bright field and GFP cubes. Total cell numbers were determined by Gen5 software using the brightfield images. Object analysis was used to count the number of mNeonGreen-positive cells in each well. The percentage of infected cells in a well was calculated as the ratio between the number of mNeonGreen positive cells and the total number of cells using the brightfield cube.

## Pseudovirus attachment assay

In microcentrifuge tubes for 1 h on ice, $2 \times 10^5$ cells were incubated with 150 μL of VSV pseudotyped with SARS-CoV-2 spike. Then, the cells were washed 3 times with chilled complete media to remove unattached viral particles. The cells were lysed in TRIzol and subjected to RNA extraction. The viral copies were measured by RT-qPCR with primers targeting VSV-N mRNA and normalized to the expression of Actin.

## Quantitative reverse transcription-PCR

As previously described [65], RNA extraction for cell samples was performed using TRI Reagent with a Direct-zol-96 RNA Kit (Zymo Research), following the manufacturer's protocol. The ImProm-II reverse transcriptase system (Promega) was used with random hexamers and 5 μL of extracted RNA to synthesize cDNA. qPCR assays for VSV were performed using SYBR dye and primers targeting VSV-N (primer 1: 5′-TGTCTACCAAGGCCTCAAATC-3′; primer 2: 5′-GTGTTCTGCCCACTCTGTATAA-3′). Predesigned PrimeTime qPCR assays (IDT) were used to quantify expression of human genes: *ACTB* (Hs.PT.39a.22214847), *LRRC15* (Hs.PT.58.26559170), *MX1* (Hs.PT.58.26787898), and *IFI44* (Hs.PT.58.21412074). A

standard curve was used to determine absolute gene copy. qPCR results were normalized to the housekeeping gene *ACTB*.

## Microscopic analysis

In 8-well chamber slides (Nunc), $4 \times 10^4$ HeLa-ACE2/HeLa-ACE2-sgLRRC15/ HeLa-ACE2-sgCD45 cells or $8 \times 10^3$ HeLa-ACE2 cells and $3.2 \times 10^4$ HeLa/HeLa-sgLRRC15 cells (1:4 ratio) were plated per well. The following day, cells were inoculated with 75 μL of SARS-CoV-2 pseudovirus for 1 h on ice. Cells were washed 3 times with chilled media to remove unattached viral particles and placed back to 37°C, 5% $CO_2$ to allow internalization. At 0 or 60 min after internalization, cells were fixed by incubation of 4% paraformaldehyde in PBS for 15 min at room temperature and permeabilized with 0.1% Triton X-100 for 10 min. Cells were subsequently incubated with recombinant anti-LRRC15 antibody (Abcam, #ab150376) and SARS--CoV-2 spike antibody [1A9] (GeneTex, #GTX632604) at 1:100, followed by incubation with 1:500-diluted Alexa Fluor 555–conjugated goat anti-rabbit IgG antibody (Abcam, #ab150078), 1:200-diluted FITC-conjugated Rat anti-mouse IgG1 antibody (BD, #553443), and phalloidin-iFluor 647 reagent (Abcam, #176759) for 60 min at room temperature. The slides were then mounted with ProLong glass antifade mountant with NucBlue staining. The fluorescence images were recorded using an Olympus FV3000 confocal microscope. The specks of spike protein on the cells were manually calculated.

To identify the protein expression of ACE2, LRRC15, DC-SIGN, the cells were incubated with purified human ACE2 (BioLegend, #375801), LRRC15 (Abcam, #ab150376), and CD209 (Biolegend, #330102) antibodies. Then, staining was done using goat anti-rat IgG, Alexa Fluor Plus 555 (Thermo Fisher, #A48263), goat anti-rabbit IgG, Alexa Fluor Plus 488 (Thermo Fisher, #A32731TR) to localize these protein expressions.

## Analysis of human lung scRNA-seq datasets

Human lung scRNA-seq datasets from non-COVID-19 patients were accessed from the Human Lung Cell Atlas (https://github.com/krasnowlab/HLCA) (Synapse #syn21041850) and from the Tissue Stability Cell Atlas (https://www.tissuestabilitycellatlas.org/) (PRJEB31843). Preprocessed R objects were downloaded from the respective repositories and utilized for analysis of cell type–specific expression patterns using Seurat [66].

Human lung scRNA-seq datasets from deceased COVID-19 patients and non-COVID-19 controls were accessed from the Single Cell Portal of the Broad Institute. We downloaded the preprocessed data from Melms and colleagues [43] (SCP1219) and Delorey and colleagues [44] (SCP1052). For both datasets, we used Seurat to filter out all annotated doublets prior to investigating cell type–specific expression patterns. We also filtered out cells from non-COVID-19 patients for visualization of *ACE2* and *LRRC15* expression. For the Melms dataset, we also compared the relative proportions of each of the fibroblast subpopulations among total fibroblasts, using a two-tailed unpaired Mann–Whitney test to assess statistical significance. On a sample-by-sample basis, we further quantified the number of *LRRC15*-expressing fibroblasts and endothelial cells (cells annotated by the authors as "Fibroblasts," or "Endothelial cells" in the [cell_type_main] slot), and *ACE2*-expressing epithelial cells (annotated as "Epithelial cells"). We then compared the relative cell frequencies in a paired manner by two-tailed Wilcoxon signed rank test.

## Analysis of bulk RNA-seq datasets

The GTEx project was supported by the Common Fund of the Office of the Director of the National Institutes of Health and by NCI, NHGRI, NHLBI, NIDA, NIMH, and NINDS

[42,67]. Bulk RNA-seq normalized TPM matrices were accessed from the GTEx Portal (https://gtexportal.org/home/datasets) on March 18, 2020, release v8 and subsequently filtered to lung samples only. Gene expression data used in this study are publicly available on the web portal and have been deidentified. Detailed clinical annotations of the GTEx cohort were obtained as controlled access data through dbGaP (phs000424.v8.p2).

To test the association between various clinical factors and *LRRC15* expression in the GTEx cohort, we employed a similar approach as previously described [68], constructing a multivariable linear regression model using age, sex, diabetes (type or type 2), hypertension, BMI, smoking, and ventilator status at time of death as predictor variables for log-transformed *LRRC15* expression. The resulting regression estimates were visualized as forest plots with 95% confidence intervals. To visualize *LRRC15* expression values after adjustment for all other clinical covariates except ventilator status, we summed the intercept and residuals from another multivariable linear regression model, omitting ventilator status as a predictor variable.

For analysis of genes correlated with *LRRC15* expression in the GTEx dataset, we computed the Spearman correlation between *LRRC15* and all other genes represented in the GTEx dataset. We performed multiple-hypothesis correction by the Benjamini–Hochberg method. For direct comparison of *ACE2* and *LRRC15* expression, we utilized log-transformed expression values to compute the linear regression line with 95% confidence intervals.

To compare *LRRC15* expression in the lungs of deceased COVID-19 patients with high versus low viral load [46], we accessed the raw count data from GSE150316 and performed differential expression analysis using DESeq2 [69]. We performed multiple-hypothesis correction by the Benjamini–Hochberg method.

To compare LRRC15 expression in lung cell lines infected with SARS-CoV-2 versus mock controls, we accessed the raw count data from GSE147507 and performed differential expression analysis using DESeq2 as above. For visualization purposes, we further extracted the normalized expression values for *LRRC15* in each of the samples and tested for statistical significance by two-tailed unpaired *t* test.

## *Trans*-inhibition assay

In 96-well plates, $4 \times 10^3$ HeLa-ACE2 cells and $1.6 \times 10^4$ HeLa/HeLa-sgLRRC15/HeLa-sgCD209 cells (1:4 ratio) were coplated per well. For 1:1 ratio coculture, $1 \times 10^4$ HeLa-ACE2 cells and $1 \times 10^4$ HeLa/HeLa-sgLRRC15/HeLa-sgCD209 cells were plated per well. The following day, media was removed from the cells and 50 μL of pseudotyped VSV were added. After incubating 1 h at 37°C, 5% $CO_2$, virus-containing media was removed and the cells were incubated in complete media for 20 h at 37°C, 5% $CO_2$. Cells were washed once and resuspended with FACS buffer, and GFP-positive cells were measured by flow cytometry using FACSCelesta (BD Biosciences). To compare GFP expressions in ACE2- and LRRC15-positive cells, pseudovirus-infected cells were stained for surface ACE2 and LRRC15 as described above with following secondary antibodies: AF405-labeled donkey anti-goat IgG (Invitrogen, #A48259) and PE-labeled donkey anti-rabbit IgG (Jackson ImmunoResearch, #711-116-152).

## Interferon stimulation

In 24-well plates, $3 \times 10^5$ A375 cells were plated per well. The following day, Universal IFN-I (R&D Systems, #11200–1) or IFN-λ2 (R&D Systems, #8417-IL-025/CF) was added at 100 U/mL or 100 ng/mL, respectively, and incubated at 37°C, 5% $CO_2$. After 6-h incubation, cells were harvested and subjected to RNA extraction and qPCR analysis as described above.

## Statistical analysis

Statistical significance was determined using GraphPad Prism 9 software. Experiments were analyzed by one-way or two-way ANOVA with Dunnett multiple comparisons test or unpaired two-pair *t* test as indicated.

## Supporting information

**S1 Data. sgRNA information and summarized results of the *surface*ome CRISPRa screen.** List of sgRNAs included in the *surface*ome CRISPRa library, raw read counts, and summarized analysis of the screen are included. Raw sequence data are deposited to NCBI's SRA database with accession number SRP349409. CRISPRa, CRISPR activation; sgRNA, single guide RNA; SRA, Sequence Read Archive.
(XLSX)

**S2 Data. The individual numerical values for the following figure panels: Figs 5A-5E, S5A, S5B and S5E–S5H.**
(XLSX)

**S3 Data. The individual numerical values for the following figure panels: Figs 2C, 2E, 3A-3G, 4A, 4B, 4D–4G, 6B, 6C, 6E, S2C–S2E, S3A, S3B, S3D, S3H, S3I, S4A, S4C and S6B–S6D.**
(XLSX)

**S1 Fig. A focused CRISPRa library was designed to induce surface proteins located on cellular plasma membrane.** (**A**) Venn diagram shows the composition of the *surface*ome CRISPRa library, which was made by integrating 4 datasets for plasma membrane proteins [59–62]. (**B**) Volcano plot showing sgRNAs targeting spike attachment factors in cells binding with SARS-CoV-2 spike S1-Fc. (**C**) Volcano plot showing sgRNAs enriched or depleted in cells sorted after staining with anti-CD45. For underlying data, see S1 Data. CRISPRa, CRISPR activation; SARS-CoV-2, Severe Acute Respiratory Syndrome Coronavirus 2; sgRNA, single guide RNA.
(TIF)

**S2 Fig. LRRC15 binds with spike proteins of SARS-CoV-2 variants and SARS-CoV-1.** (**A, B**) A375 cells (**A**) or HeLa cells (**B**) were transduced with indicated activating sgRNAs and surface expression of LRRC15 and ACE2 was measured by flow cytometry. (**C**) HeLa cells were transduced with indicated activating sgRNAs and incubated with 100, 500, or 1,000 μg/mL full-length SARS-CoV-2 spike protein. Protein binding was measured by flow cytometry and calculated as MFI. (**D**) The binding of ACE2 or LRRC15 to immobilized histidine-tagged spike proteins from SARS-CoV-2 mutant variants was measured using ELISA assay. (**E**) The binding affinity of ACE2 or LRRC15 to SARS-CoV-2, SARS-CoV-1, and MERS-CoV spike protein RBD fragment was determined by ELISA. For underlying data, see S3 Data. ACE2, angiotensin-converting enzyme 2; LRRC15, leucin-rich repeat-containing 15; MERS-CoV, Middle East Respiratory Syndrome Coronavirus; MFI, mean fluorescence intensity; RBD, receptor-binding domain; SARS-CoV-1, Severe Acute Respiratory Syndrome Coronavirus; SARS-CoV-2, Severe Acute Respiratory Syndrome Coronavirus 2; sgRNA, single guide RNA.
(TIF)

**S3 Fig. LRRC15 is not an entry receptor for SARS-CoV-2.** (**A, B**) A375 cells (**A**) or HeLa cells (**B**) transduced with indicated activating sgRNAs were infected with VSV pseudoviruses VSVΔG-S-SARS2 or VSVΔG-G. GFP signal was measured at 20 hpi by flow cytometry (*n* = 3). (**C**) HeLa cells stably expressing ACE2 (HeLa-ACE2) were assessed for cell surface expression

of ACE2 by flow cytometry. (**D**) HeLa or HeLa-ACE2 cells were infected with VSV pseudo-virus harboring SARS-CoV-2 spike and GFP signal was measured at 20 hpi by flow cytometry ($n$ = 4). (**E**) HeLa-ACE2 cells transduced with indicated activating sgRNAs were assessed for surface expression of LRRC15 by flow cytometry. (**F**) HeLa-ACE2 cells expressing LRRC15 or empty vector were assessed for surface expression of LRRC15 by flow cytometry. (**G**) Huh7.5 cells transduced with indicated activating sgRNAs were assessed for surface expression of LRRC15 by flow cytometry. (**H**, **I**) HeLa-ACE2 cells transduced with indicated activating sgRNAs (**H**) or 293T-ACE2/TMPRSS2 cells expressing LRRC15 or empty vector (**I**) were pre-treated with DMSO or camostat for 1 h and infected with VSVΔG-S-SARS2. GFP signal was measured at 20 hpi by flow cytometry ($n$ = 3). Data represent means ± SD (**A**, **B**, **D**, **H**, **I**). Data were analyzed unpaired two-tailed $t$ test (**D**, **H**, **I**); ns, not significant; $^*p < 0.05$; $^{**}p < 0.01$. For underlying data, see **S3 Data**. ACE2, angiotensin-converting enzyme 2; GFP, green fluorescent protein; hpi, hours postinfection; LRRC15, leucin-rich repeat-containing 15; SARS-CoV-2, Severe Acute Respiratory Syndrome Coronavirus 2; sgRNA, single guide RNA; TMPRSS2, transmembrane protease serine 2; VSV, vesicular stomatitis virus.
(TIF)

**S4 Fig. LRRC15 does not interact with ACE2.** (**A**) Huh7.5 cells were transduced with indicated activating sgRNAs. Cell surface expression of ACE2 was measured by flow cytometry and calculated as MFI. Data represent means ± SD ($n$ = 3). (**B**) Representative images of immunofluorescence staining of LRRC15 or DC-SIGN (green), ACE2 (red), Actin (blue), and DAPI (cyan) in HeLa-ACE2 cells transduced with indicated activating sgRNAs. (**C**) Binding of Fc-tagged recombinant human ACE2 to His-tagged LRRC15 was measured by ELISA. SARS-CoV-2 spike protein was used as a positive control, MERS-spike protein as a negative control. (**D**) Binding of SARS-CoV-2 spike S1-Fc to HeLa-ACE2 cells transduced with indicated activating sgRNAs or a LRRC15-expressing vector was measured by flow cytometry. For underlying data, see **S3 Data**. ACE2, angiotensin-converting enzyme 2; LRRC15, leucin-rich repeat-containing 15; MERS, Middle East Respiratory Syndrome; MFI, mean fluorescence intensity; SARS-CoV-2, Severe Acute Respiratory Syndrome Coronavirus 2; sgRNA, single guide RNA.
(TIF)

**S5 Fig. LRRC15 expression in the human lung is cell type specific and inversely associated with ventilator usage.** (**A**, **B**) Cell type–specific expression of ACE2 and LRRC15, assessed by scRNA-seq of human lungs from (**A**) the Human Lung Cell Atlas [41] or (**B**) the Tissue Stability Cell Atlas [42]. (**C**) Forest plot of the multivariable regression coefficients for various clinical features as predictors for lung expression of LRRC15 (from GTEx; $n$ = 554 samples) [43]. (**D**) Tukey boxplots (IQR boxes with $1.5 \times$ IQR whiskers) showing LRRC15 expression after adjustment for clinical features noted in (**C**), classifying each lung sample by ventilator status at the time of death. Statistical significance was assessed by two-tailed unpaired Mann–Whitney test. (**E**) Spearman correlation analysis between LRRC15 and all other genes, based on RNA-seq of human lung samples from GTEx. Multiple hypothesis correction was performed by the Benjamini–Hochberg method. (**F**) Scatter plot comparing log-normalized expression of LRRC15 and ACE2 across human lung samples. Spearman rho = −0.343, $p = 2.05 ^* 10^{-17}$. The linear regression line is overlaid, with 95% confidence intervals shaded in. (**G**) Cell type–specific expression of genes that are highly correlated with LRRC15, assessed by scRNA-seq of human lungs from the Tissue Stability Cell Atlas. (**H**) Frequency of LRRC15-expressing fibroblasts and endothelial cells and ACE2-expressing epithelial cells were assessed by scRNA-seq of human lungs and presented in Tukey boxplots (IQR boxes with $1.5 \times$ IQR whiskers). For underlying data, see **S2 Data**. ACE2, angiotensin-converting enzyme 2; GTEx, Genotype-Tissue Expression; IQR, interquartile range; LRRC15, leucin-rich repeat-containing 15; scRNA-

seq, single-cell RNA-sequencing.
(TIF)

**S6 Fig. LRRC15 inhibits ACE2-mediated SARS-CoV-2 entry *in trans*.** (**A**) Left, surface ACE2 and LRRC15 expressions of HeLa, HeLa-ACE2, or HeLa-sgLRRC15 cells were measured by flow cytometry. Right, after coculturing HeLa-ACE2 cells with HeLa-sgLRRC15 cells and infecting with VSVΔG-S-SARS2, surface ACE2 and LRRC15 expressions were measured by flow cytometry. GFP expressions were compared in LRRC15+ (HeLa-sgLRRC15) cells and ACE2+ (HeLa-ACE2) cells. (**B**) HeLa-ACE2 cells were cocultured with HeLa, HeLa-sgLRRC15, or HeLa-sgCD209 cells at 1:1 ratio and infected with VSV pseudovirus–harboring spike of SARS-CoV-2 Wuhan-hu-1 strain at high or low titer. GFP signal was measured at 20 hpi by flow cytometry ($n = 4$). Representative of 3 independent experiments are shown. (**C**) *Trans*-inhibition assay was performed as in (**B**) with VSV pseudovirus–harboring spike of SARS-CoV-2 Delta variant (B.1.617.2 strain) ($n = 4$). Representative of 3 independent experiments are shown. (**D**) A375 cells were treated with Universal IFN-I (100 U/mL) or IFN-λ2 (100 ng/mL) and harvested after 6 h. RNA expressions of LRRC15, IFI44, and MX1 was measured by qPCR and normalized to ACTB. Data represent means ± SD and were analyzed by one-way ANOVA with Dunnett multiple comparisons test (**B**, **D**) or unpaired two-tailed $t$ test (**C**). ns, not significant; $^*p < 0.05$; $^{**}p < 0.01$; $^{***}p < 0.001$; $^{****}p < 0.0001$. For underlying data, see **S3 Data**. ACE2, angiotensin-converting enzyme 2; GFP, green fluorescent protein; hpi, hours postinfection; LRRC15, leucin-rich repeat-containing 15; SARS-CoV-2, Severe Acute Respiratory Syndrome Coronavirus 2; VSV, vesicular stomatitis virus.
(TIF)

**S7 Fig. Gating strategy for flow cytometry analysis.** Cells were infected with VSV pseudovirus–harboring spike of SARS-CoV-2, and GFP signal was measured by flow cytometry. Representative datasets for HeLa-ACE2, HeLa-ACE2-sgLRRC15 cells, and uninfected cells are shown. GFP, green fluorescent protein; SARS-CoV-2, Severe Acute Respiratory Syndrome Coronavirus 2; VSV, vesicular stomatitis virus.
(TIF)

## Acknowledgments

We thank Robert Orchard for manuscript review and discussion. We also thank Megan Baldridge, Rachel Rodgers, Leran Wang, and William Buchser for helping to establish the bioinformatic analysis pipeline. We acknowledge the Brown University Flow Cytometry and Sorting Facility, the Genomics facility, the Leduc Bioimaging facility, and the Lentivirus Core for help with critical analysis.

## Author Contributions

**Conceptualization:** Jaewon Song, Sanghyun Lee.

**Data curation:** Jaewon Song, Ryan D. Chow, Li Zhang, Sanghyun Lee.

**Formal analysis:** Jaewon Song, Li Zhang, Sanghyun Lee.

**Funding acquisition:** Olin D. Liang, Craig B. Wilen, Sanghyun Lee.

**Investigation:** Sanghyun Lee.

**Methodology:** Ryan D. Chow, Mario A. Peña-Hernández, Li Zhang, Skylar A. Loeb, Eui-Young So, Olin D. Liang, Ping Ren, Sidi Chen, Sanghyun Lee.

**Project administration:** Sanghyun Lee.

**Resources:** Eui-Young So, Olin D. Liang, Ping Ren, Sidi Chen, Craig B. Wilen, Sanghyun Lee.

**Supervision:** Craig B. Wilen, Sanghyun Lee.

**Validation:** Jaewon Song, Mario A. Peña-Hernández, Li Zhang, Skylar A. Loeb, Eui-Young So, Craig B. Wilen, Sanghyun Lee.

**Visualization:** Jaewon Song, Ryan D. Chow, Li Zhang, Sanghyun Lee.

**Writing – original draft:** Jaewon Song, Ryan D. Chow, Skylar A. Loeb, Sanghyun Lee.

**Writing – review & editing:** Jaewon Song, Ryan D. Chow, Mario A. Peña-Hernández, Li Zhang, Skylar A. Loeb, Ping Ren, Sidi Chen, Sanghyun Lee.

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
