## [Editor Report · Decision Letter 0]

6 Dec 2021

Dear Dr. Lee, 

Thank you for submitting your manuscript entitled "LRRC15 is an inhibitory receptor blocking SARS-CoV-2 spike-mediated entry in trans" for consideration as a Research Article by PLOS Biology.

Your manuscript has now been evaluated by the PLOS Biology editorial staff, as well as by an academic editor with relevant expertise, and I am writing to let you know that we would like to send your submission out for external peer review.

Once your full submission is complete, your paper will undergo a series of checks in preparation for peer review. Once your manuscript has passed the checks it will be sent out for review. To provide the metadata for your submission, please Login to Editorial Manager (https://www.editorialmanager.com/pbiology) within two working days, i.e. by Dec 08 2021 11:59PM.

If your manuscript has been previously reviewed at another journal, PLOS Biology is willing to work with those reviews in order to avoid re-starting the process. Submission of the previous reviews is entirely optional and our ability to use them effectively will depend on the willingness of the previous journal to confirm the content of the reports and share the reviewer identities. Please note that we reserve the right to invite additional reviewers if we consider that additional/independent reviewers are needed, although we aim to avoid this as far as possible. In our experience, working with previous reviews does save time. 

If you would like to send previous reviewer reports to us, please email me at pjaureguionieva@plos.org to let me know, including the name of the previous journal and the manuscript ID the study was given, as well as attaching a point-by-point response to reviewers that details how you have or plan to address the reviewers' concerns. 

Given the disruptions resulting from the ongoing COVID-19 pandemic, please expect some delays in the editorial process. We apologise in advance for any inconvenience caused and will do our best to minimize impact as far as possible.

Kind regards,

Paula

Paula Jauregui, PhD

Editor

PLOS Biology

---

## [Decision Letter · Decision Letter 1]

9 Feb 2022

Dear Dr Lee,

Thank you very much for submitting your manuscript "LRRC15 is an inhibitory receptor blocking SARS-CoV-2 spike-mediated entry in trans" for consideration as a Research Article at PLOS Biology. Your manuscript has been evaluated by the PLOS Biology editors, an Academic Editor with relevant expertise, and by independent reviewers.

As you will see in the reviews attached below, the reviewers find your manuscript interesting and are overall positive about the novelty of your findings, but they also raise serious overlapping concerns regarding the physiological relevance of the LRRC15 interaction and the strength of the conclusions that can be drawn from your data. It would therefore be critical that you confirm experimentally whether your findings apply in the context of natural SARS-CoV-2 infections in more physiologically-relevant cells in which LRRC15 proteins are expressed at endogenous levels. Please also ensure that you address any discrepancies between your findings and any existing results on the topic.

In light of the reviews, we would be open to inviting a revision of your manuscript that fully addresses the reviewers' comments. Please note that we cannot make any decision about publication until we have seen the revised manuscript and your response to the reviewers' reports, and your revised manuscript is likely to be sent for further evaluation by the reviewers.

We expect to receive your revised manuscript within 6 months.

**IMPORTANT - SUBMITTING YOUR REVISION**

*Resubmission Checklist*

*Published Peer Review*

*PLOS Data Policy*

*Blot and Gel Data Policy*

Sincerely,

Dario

Dario Ummarino, PhD

Senior Editor

PLOS Biology

dummarino@plos.org

REVIEWS:

Reviewer #1: 

This submission demonstrates that LRRC15 is an integral plasma membrane protein that binds SARS CoV spikes unproductively, in doing so, titrating viruses from ACE2 receptor-mediated virus entry and infection, at least in those conditions where the LRRC15 is overexpressed. The data are solid, although at times not especially striking, and show that the LRRC15 titration effect can be in "cis" (LRRC15 on same cells as ACE2) or in "trans" (LRRC15 not on the same cells as ACE2). The sequestration of virus onto LRRC15 has a modest antiviral effect. Authors make suggestions about the relevance of LRRC15 in human COVID19, but it remains unclear whether LRRC15 in vivo is sufficiently abundant to absorb viruses and keep them from infecting through ACE2 dependent entry pathways. 

The strengths of the report are with its convincing discovery of LRRC15 as a spike binding protein, the demonstration that LRRC15 binds spike RBDs but does not sterically interfere with ACE2, and with the data that sufficient overexpression of LRRC15 can have some antiviral activity. The importance of the report comes with the clear identification of a SARS-2 spike binding protein on host cells. It is important to know about the cells that bind SARS-CoV-2 virus particles. Of note, there have been several reports of host proteins that attach SARS-2 particles to cells; this submission stands out with strong data that confidently show that LRRC15 binds SARS-2 spikes at low affinity. 

There are some concerns with the submission.

1. A central concern is whether the positive results arise only if LRRC15 is vastly overexpressed. Some activating sgRNAs generate LRRC15 levels exceeding those obtained by direct plasmid based LRRC15 overexpression (Fig. S2 and S3). At these high overexpression levels, only very modest levels of SARS-2 spikes bind cells (Fig 2). It takes very high LRRC15 expression to interfere with SARS-2 pseudovirus entry (Fig 3). Similarly, it takes LRRC15 overexpression to witness trans-inhibition of virus entry (Fig. 6). It makes sense that vast LRRC15 overexpression is necessary to obtain the cell culture assay readouts, because LRRC15 binds SARS-2 spikes with 200 times lower affinity than ACE2 (for SARS-1 spikes, LRRC15 binding is 1000 times less strong than ACE2). It may be difficult to compare LRRC15 surface levels in the cell culture experiments to the physiological levels in lung fibroblasts and endothelial cells, but such comparisons would help in determining whether the findings in the paper have physiological relevance. Comparison to physiologic LRRC15 levels is recommended.

2. A second concern is with the important data in Fig 6; which is aimed at supporting the key conclusion that LRRC15 has "trans-inhibitory" activity. The spike and LRRC15 signals are very hard to see in Fig 6D and in part this may be because there are no stains for plasma membrane. Additional PM staining and better imaging of virus-LRRC15 coincidence is recommended. Also, it is the ratio of cell surface ACE2 to LRRC15 that is relevant, and this could be estimated, as the question is whether it requires hundreds-fold more LRRC15 than ACE2 to effect virus inhibition, given that the virus affinity to LRRC15 is in the 0.1 to 0.2 mM range.

3. In virology, the term "receptor" is used for a surface protein that does more than simply bind viruses. Bona-fide receptors both bind viruses and induce changes in virus that allow for successful virus entry and genome delivery. By these definitions, LRRC15 is not a receptor. It is more appropriate to describe LRRC15 as a virus attachment factor (an attachment factor that can interfere with ACE2 receptor-driven entry), not a receptor. 

Minor points:

1. Lines 58-60; RBD of SARS-CoV-2 breaks murine host barrier only if there are substitutions such as N501Y; consider rephrasing lines 59-60.

2. Lines 183-184; unclear what is meant here; is the suggestion that membrane-anchored LRRC15 at high density on cell surfaces allows for multivalent avidity to viruses, while soluble LRRC15 binding to virus relies only on affinity, not avidity? Also, lines 185-187; data do not appear to allow any conclusions about LRRC15 "restricting the internalization" of virions.

3. Fig. S2D; Kd for SARS-1 RBD to ACE2 is 0.16 nM, higher affinity than SARS-2 RBD. This is not consistent with many reports that SARS-2 RBD has the higher affinity for ACE2.

4. Line 338; not clear whether the trans-inhibition in HeLa cell context "indicates that the antiviral effect of LRRC15 can be broad in a physiological context". Consider tempering conclusions here. 

Reviewer #2 (Teunis BH Geijtenbeek): Here the authors have identified LRRC15 as an attachment receptor for SARS-CoV-2 that does not by itself mediate entry. Co-expression with ACE2 on cell-lines strongly suggest that LRRC15 suppresses ACE2-mediated infection of SARS-CoV-1. Using public data sets they show that LRRC15 is not expressed by ACE2-positive cells but rather by lymphatic endothelial cells. They have identified pathological fibroblasts as a major cell type expressing LRRC15 which are present in lungs of deceased SARS-CoV-2 patients. They suggest that LRRC15 has an antiviral function as high expression correlates with low viral load in deceased patients. Finally they show that LRRC15 can inhibit virus infection in trans. 

The identification of LRRC15 as an attachment receptor is interesting and the inhibitory in trans function is novel but lacks proper controls to convincingly show it is a novel function. The authors attempt to find a physiological role for LRRC15 in data sets from lungs of deceased SARS-COV-2 Patients but the conclusions are overstated as no real correlation can be made without extensive testing. All patient data are based on transcriptomics/RNA seq but protein expression is not investigated. Moreover, in order to exert the putative inhibitor function the cells need to be localized next to each other. The localization of the cells compared to ACE2 cells needs to be investigated. Finally, everything has been done with pseudotyped viruses and cell-lines, which lowers relevance. SARS-CoV-2 isolates need to be used in binding and infection experiments to understand whether LRRC15 is able to inhibit actual virus. Coronaviruses express high levels of Spike protein on membrane, whereas Spike on pseudoviruses will be much less and this has a major impact on the findings. 

Major concerns

- Only pseudotyped virus is used and the major experiments need to be performed with SARS-CoV-2 isolates. This is essential as the Spike proteins are much higher on actual virus than pseudovirus. 

- The LRRC15 binding to S protein is very weak (Fig 2). These data should be quantified. Here both SARS-CoV-2 and pseudotyped virus can be used to measure binding as multivalency might increase interaction (similar as has been done for Fig 4). 

- Induction of LRRC15 on ACE2 cells inhibits pseudovirus infection (Fig 3). The data are relative and actual infection data for all relative data should be shown to understand the impact on infection. 

- There is a small effect of LRRC15 induction but not overexpression on ACE2 expression and this should be quantified to understand relevance. What would happen if LRRC15 was induced on epithelial cells or CALU-3 cells that have endogenous ACE2 expression? 

- Fig 4B. The authors state that virus is sequestered by LRRC15 on ACE2-positive cells but here they should also show ACE2-negative cells as it is possible that the increase is due to LRRC15 and virus binding. 

Based on these data (Fig 4B) the authors suggest that LRRC15 sequesters virus on cell-surface and prevents internalisation (line 174). But this cannot be concluded from an experiment performed on ice. The authors need to show that the virus in presence of LRRC15/ACE2 is not internalized compared to ACE2 alone. For example by trypsin removal of extracellular bound virus and pulse chase experiments or tracking by immunostainings. 

The possible association of LRRC15 expression with deceased SARS-CoV-2 patients (SFig 5, Fig5) is very suggestive and not convincing. The pathophysiological role is highly speculative and not relevant as it is unclear how expression of lRRC15 is regulated. Further experiments would be required such as co-localization of LRRC15 positive cells and epithelial cells, and actual infection of epithelial cells and expression of LRRC15 (Fig 5E). Without more concrete data it is very difficult to convince that LRRC15 has an antiviral function in vivo. The low virus load could be due to many different processes as also shown by the high amount of differently expressed genes and as such it is an overstatement to state that LRRC15 might contribute to the observed decrease of virus load. 

Fig 6. infectivity is very low 2% or 0.3%. Why is the infection so low and this makes the effect although significant also less relevant. Moreover, controls need to be included to show that the co-culture with just any attachment receptor does not prevent infection. So ACE2-positive cells should be co-cultured with DC-SIGN-positive cells (which have shown to enhance infection in trans/cis). Is a ratio of 1:4 physiologically relevant? It is likely that LRCC15 sequesters virus and thereby lowers virus concentration in the medium leading to lower infection so this needs to be carefully controlled. 

The colocalization data are very unclear and it is difficult to assess co-localization. significance should be denoted in Fig 6E after quantification. 

Reviewer #3: In this manuscript, Dr. Wilen and Lee's groups have used a genome-wide CRISPR activation screening to determine novel SARS-CoV-2 spike binding partners. They used ACE2 as a positive control for their screens, and identified LRRC15 as a new interaction partner specifically for the spike protein of SARS2/various SARS2 variants, as well as SARS1. A serial of experiments were also performed to verify this interaction, and the binding epitope was mapped to RBD, although with much weaker affinity compared to ACE2 (~100 fold less by ELISA assay). Functionally, LRRC15 itself does not permit viral entry as indicated by VSV-G SARS-2 spike pseudotyped virus infection experiments. However, LRRC15 provided in cis (on the same ACE2 positive cells, such as A375 melanoma or Hela cells) or trans (LRRC15+ cells plus ACE2+ cells) significantly inhibited SARS-2 spike pseudovirus infection. Mechanistically, the authors provided some evidence that LRRC15 may capture SARS-2 spike pseudovirus and potentially render viron sequestration. Given the fact that LRRC15 showed distinct expression pattern compared to ACE2, the author proposed that LRRC15 provided in trans on SARS-2 non-permissive fibroblasts may protect ACE2 permissive cells against viral infection, and LRRC15 is an inhibitory molecule for SARS2 viral entry by directly interacting with the spike protein. 

Overall, this study contains novel findings and is quite interesting to the field. The paper was also well-written. However, there are several technical concerns and a lack of detail mechanisms which I see for potential improvement.

1. Since this is a genome-wide CRISPRa screening, did the authors identify any other molecules besides LRRC15 from this screen? There are around 10 other SARS-2 binding partners (C-type lectins, AXL, Siglec, Kremen-1 etc,) being identified through various screening platforms, the comparison of CRISPRa with other methods will provide important technical insights to the field. Moreover, the authors may discuss the advantage and disadvantages of CRISPRa screen vs others in this specific experimental setting.

2. In figure 2, the LRRC15 binding data look quite weak in general. However, it seems specific although the Kd of this interaction is ~200 fold lower based the direct protein-protein interaction measurement by ELISA. Other than RBD, does CTD or S2 contribute to LRRC15 binding? Moreover, the difference between Spike binding to ACE2 and LRRC15 seems much smaller in Delta or Iota variants, the authors may need to discuss whether there is any meaning interpretation on these interesting findings.

3. Using SARS-2 spike VSV-G pseudovirus, the authors found LRRC15 itself does not elicit pseudovirus infection, which is different from VSV-G spike pseudovirus. Coexpression of both LRRC15 and ACE2 significantly inhibited SARS2 pseudovirus infection. This data is interesting, however, requires delicate validation with active SARS-2 virus. The infection mechanism of VSV-G pseudovirus has been found to be quite different from the real SARS-2 virus (PMCID: PMC8106883). For instance, DC-SIGN/L-SIGN can elicit pseudovirus infection, but these molecules did not elicit active replication of SARS2 virus. The authors may also need to include DC-SIGN as a control to show whether their pseudovirus can infect receptors other than ACE2, and whether LRRC15 in cis could inhibit ACE2 dependent SARS2 real virus infection. 

4. Mechanistically, the authors did not find LRRC15 can modulate ACE2. In figure 4B, LRRC15 CRISPRa induction on the top of ACE2 was found to increase viron binding. However, a LRRC15 only control is required for this experiment. The authors found LRRC15 fusion protein can partially block pseudovirus binding (Fig 4C), in a much weaker fashion than ACE2, which could be largely attributed to the weak affinity of LRRC15 to spike RBD compared to ACE2.Therefore, the authors reasoned that "the inhibitory function of LRRC15 may require localization to the cellular membrane and/or the cytosolic domain of LRRC15 for its functional efficiency", which is not convincing in this case. Moreover, LRRC15 fusion could not block ACE2-spike binding, which is not enough to conclude that LRRC15 could not compete with ACE2, particularly in cis. Moreover, does ACE2 fusion protein block LRRC15/RBD binding? This is important and will provide important information that whether LRRC15 may share some similar epitopes with ACE2 within RBD.

5. The authors only showed that LRRC15 enhanced pseudovirus binding in the presence of ACE2. Does LRRC15 elicit direct virus endocytosis and degradation (no replication), in the condition with or without ACE2? This is a valid hypothesis that requires detail characterization, particularly in the absence of ACE2.

6. The LRRC15 expression data is rather interesting, which indicates LRRC15 expression in ACE2 negative cells. Importantly, LRRC15 provided in trans also inhibited SARS2 pseudovirus infection. Again, this experiments require further validation using real virus. Given the fact that the C-type lectins in trans can enhance SARS2 infection, the authors may need to include DC-SIGN as a control for this experiment, and measure the endocytosis of virus by LRRC15/DC-SIGN/ACE2, to pinpoint specific mechanisms.

7. TMPRSS2 has been suggested to be critical to SARS2 infection, what is the role of LRRC15 in the presence of this protease?

---

## [Decision Letter · Decision Letter 2]

17 Aug 2022

Dear Dr Lee,

Thank you for your patience while we considered your revised manuscript "LRRC15 is an inhibitory attachment factor blocking SARS-CoV-2 entry in trans" for publication as a Research Article at PLOS Biology. I have taken over the handling of your submission during my colleague Paula Jauregui's absence from the office this week, in order to prevent any unnecessary loss of time. 

This revised version of your manuscript has been evaluated by the PLOS Biology editors, the Academic Editor and the original reviewers, one of whom -Teunis Geijtenbeek (R2)- has signed his report. Based on the reviews and our Academic Editor's assessment of your revision, we are very likely to accept this manuscript for publication, provided you satisfactorily address the remaining points raised by the reviewers and any issues related to data availability, reporting and other policy-related requests.

R1's remaining concern is similar to comment #1 by R3, which is hard to address experimentally. We would thus suggest that you address it with changes to the text that emphasize the limitations of the results. Likewise, R3 comments #2 and #3 are important and the manuscript should include in the discussion section explicit statements about the limitations of the work, including rephrasing of the conclusions to address R3 comment #3. The elimination of LLRC15 effect by TMPRSS2 does not imply that LRRC15 only blocks the endosomal pathway of virus entry and this should thus be rephrased after additional consideration.

In addition, please also address the following editorial issues:

1) We would like to suggest a modification to the title, so that it reads better and is more accessible to a broad and international readership. How about "LRRC15 inhibits SARS-CoV-2 cellular entry in trans"? We feel the details of the nature of the inhibition are sufficiently explained in the abstract and this title change would draw more readers to the work.

2) Your data availability statement includes the following: "All other data are available upon reasonable request." You may be aware of the PLOS Data Policy, which requires that all data underlying the main and supplementary figures be made available without restriction: http://journals.plos.org/plosbiology/s/data-availability. For more information, please also see this editorial: http://dx.doi.org/10.1371/journal.pbio.1001797

Note that we do not require all raw data. Rather, we ask that all individual quantitative observations that underlie the data summarized in all of the figure panels and results of your paper be made available in one of the following forms:

 - Supplementary files (e.g., excel). Please ensure that all data files are uploaded as 'Supporting Information' and are invariably referred to (in the manuscript, figure legends, and the Description field when uploading your files) using the following format verbatim: S1 Data, S2 Data, etc. Multiple panels of a single or even several figures can be included as multiple sheets in one excel file that is saved using exactly the following convention: S1_Data.xlsx (using an underscore).

 - Deposition in a publicly available repository. Please also provide the accession code or a reviewer link so that we may view your data before publication. 

Regardless of the method selected, please ensure that you provide the individual numerical values that underlie the summary data displayed in all main and supplementary figure panels as they are essential for readers to assess your analysis and to reproduce it:

3) Please also ensure that all of the main and supplementary figure legends in your manuscript include information on the precise location where the underlying data can be found, be it a repository of a supplemental file. 

4) Please also ensure that your supplemental data file/s have a legend that enables readers to understand what they include. 

5) Please ensure that your Data Availability Statement in our electronic submission system accurately describes where all of your data can be found.

6) The figure legends to figures 5c, 5e, 6b, 6c, 6e and Supp 4a are missing information about what the bars represent (mean, median,...) the type of error that is plotted and, for 5c, what the box limits represent.

7) For each type of flow cytometry experiment, supplementary figure(s) need to be provided depicting the flow cytometry gating strategy used (similar to that provided in Supp Fig 6).

We expect to receive your revised manuscript within two weeks. 

*Published Peer Review History*

*Press*

With best wishes,

Nonia

Nonia Pariente, PhD

Editor in Chief

PLOS Biology

npariente@plos.org

on behalf of 

Paula

Editor,

pjaureguionieva@plos.org,

PLOS Biology

Reviewer remarks:

Reviewer #1: 

This revised submission is improved. The essential conclusions of the submission are now more adequately defended with new findings. The findings that LRRC15 suppresses authentic SARS-CoV-2 are valuable. The article is a significant contribution. Notably some of the findings are largely concordant with independently obtained results from other labs who had identified LRRC15 as a virus attachment factor.

Remaining Comment:

Can the authors please point out the IFA signals for spike in Figs 4C and 6D? The authors claim that VSV-SPIKE pseudo viruses bind LRRC15+ cells in Fig 4C but this reviewer is not confident of the signals deemed positive vs those considered negative. In Fig 4C there seem to be no positive spike signals on LRRC15 cells but they seem to be present in Fig 6D. Again, it would be helpful to point out what green flouresence is and is not deemed evidential psueudo virus binding with some arrows.

Reviewer #2 (Teunis BH Geijtenbeek): 

The authors have addressed the concerns with new data and clear explanations. The manuscript has improved and I have no other concerns.

Reviewer #3: 

The authors have reasonably answered my concerns. Some minor issues:

1.The authors commented that LRRC15 sequestrates the virus on the surface and prevents the endocytosis of the virus, which is lack of clear evidence. It remains important to test whether LRRC15 in cis or in trans can mediate virus endocytosis, in a way potentially important to reduce virus titer. The identification of virus on the cell surface is not enough to dissect this question. More experiments, for example, whether LRRC15 effect is lost in 4 degree vs 37 degree, and/or the inclusion of endocytosis inhibitors, are necessary.

2. The authors may need to include some discussion on the limitation of this study--such as the inhibitory effect of LRRC15 on virus infection mediated by ACE2 could be conditional, particularly in physiological conditions.

3. The explanation on the data related to TMPRSS2 is still not convincing. The authors found LRRC15 suppressed viral entry only upon camostat-treatment, but not with control treatment, which may indicate the role of LRRC15 can be dependent on, at least in the case of TMPRSS2 overexpression. Given the fact that TMPRSS2 is important for ACE2 priming, the author's statement that "In 293TACE2/TMPRSS2 cells, LRRC15 suppressed viral entry only upon camostat-treatment but not with control-treatment (S3I Fig), implying the endosomal pathway of SARS-CoV-2 entry is sensitive to the entry restriction by LRRC15 over plasma membrane fusion pathway" may be not accurate.

---

## [Editor Report · Decision Letter 3]

25 Aug 2022

Dear Dr Lee,

Thank you for the submission of your revised Research Article "LRRC15 inhibits SARS-CoV-2 cellular entry in trans" for publication in PLOS Biology. On behalf of my colleague Paula Jauregui (who is out of the office this week) and the Academic Editor, Tom Gallagher, I am pleased to say that we can in principle accept your manuscript for publication, provided you address any remaining formatting and reporting issues. 

Please note that you have not yet fully addressed the requirements of our data provision policy:

- Your Data Accessibility Statement (DAS) in our submission system still says that "All other data are available upon reasonable request." Please note that we cannot accept this. All of the data needs to either be deposited in publicly available repositories or contained within the supplementary information. Once all data is available, please remove this statement from the DAS.

- Every legend to every main and supplementary figure needs to indicate where the data underlying the figure can be found, be it in a supplementary file or in an external repository (ie, you can refer to data in S1 Data or in the SRA, giving the accession number that provides data for that figure). The data underlying each main and supplementary figure needs to be made available and its location stated in the DAS in general terms and in each figure legend more specifically.

In addition to these issues above, please address any issues detailed in an email you should receive within 2-3 business days from our colleagues in the journal operations team; no action is required from you until then. Please note that we will not be able to formally accept your manuscript and schedule it for publication until you have completed any requested changes.

PRESS

Sincerely, 

Nonia

Nonia Pariente, PhD

Editor in Chief

PLOS Biology 

on behalf of 

Editor

PLOS Biology
